**Algal lipids reveal unprecedented warming rates in alpine areas of SW Europe**
**during the Industrial Period**
Antonio García-Alix[1,2,3]*, Jaime L. Toney[2], Gonzalo Jiménez-Moreno[1], Carmen Pérez-
Martínez[4], Laura Jiménez[4], Marta Rodrigo-Gámiz[1], R. Scott Anderson[5], Jon Camuera[6],
Francisco J. Jiménez-Espejo[3], Dhais Peña-Angulo[7], María J. Ramos-Román[6]
[1] Department of Stratigraphy and Paleontology, University of Granada, Granada, 18072,
Spain.
[2] School of Geographical and Earth Sciences, University of Glasgow, Glasgow, G12
8QQ, UK.
[3] Instituto Andaluz de Ciencias de la Tierra (IACT), CISC-UGR, Armilla, 18100, Spain.
[4] Department of Ecology and Institute of Water Research, University of Granada,
Granada, 18072, Spain.
[5] School of Earth and Sustainability, Northern Arizona University, Flagstaff, AZ 86011,
USA.
[6] Department of Geosciences and Geography, University of Helsinki, Helsinki, FI-00014,
Finland.
[7] Department of Geography, University of Zaragoza, Zaragoza, 50009 Spain.
* *Correspondence to:* Antonio García-Alix (agalix@ugr.es)

**Abstract.** Alpine ecosystems of the southern Iberian Peninsula are among the most vulnerable and the first to respond to modern climate change in southwestern Europe. While major environmental shifts have occurred over the last ~1500 years in these alpine ecosystems, only changes in the recent centuries have led to abrupt environmental responses, but factors imposing the strongest stress have been unclear until now. To understand these environmental responses, this study, for the first time, has calibrated an algal lipid-derived temperature proxy (based on long-chain alkyl diols) to instrumental historical data extending alpine temperature reconstructions to 1500 years before present. These novel results highlight the enhanced effect of greenhouse gases on alpine temperatures during the last ~200 years and the long-term modulating role of solar forcing. This study also shows that the warming rate during the 20$^{th}$ century (~0.18ºC/decade) was double that of the last stages of the Little Ice Age (~0.09ºC/decade), even exceeding temperature trends of the high-altitude Alps during the 20$^{th}$ century. As a consequence, temperature exceeded the pre-industrial records in the 1950s, and was one of the major forcings of the enhanced recent change in the alpine ecosystems from southern Iberia. Nevertheless, other factors reducing the snow and ice albedo (e.g., atmospheric deposition) may have influenced local glacier loss, since almost steady climate conditions predominated from middle 19$^{th}$ century to the first decades of the 20$^{th}$ century.

**1. Introduction**

Global mean annual surface temperatures have risen by ~0.85ºC from 1880 to 2012 and the recent decades have been the warmest in the Northern Hemisphere during the Common Era (IPCC, 2013). This trend is alarming, since over the last decade

temperature records have been broken yearly. For example, in Spain the highest

temperatures ever recorded in September and July occurred in 2016 (45.5ºC) and 2017

(46.9ºC-47.3ºC), respectively (Spanish National Weather Agency - AEMet Open Data,

2019). Increasing global temperatures are contributing not only directly to land and ocean

surface warming, but are also changing the global hydrological cycle through the

disturbance of atmospheric circulation patterns and moisture (Easterling et al., 2000;

IPCC, 2013). As a result, the term "global warming" is migrating towards recent "climate

change" in order to express the variety of modern climate extremes witnessed across the

world. The effects of modern global warming and associated climate change events may

be causing extreme environmental impacts, beyond what is recorded in the recent

geologic record (Waters et al., 2016). Hence, it is crucial to identify warming thresholds,

rates, and forcing mechanisms from past high-resolution temperature records to

understand modern climate change. It is especially important in fragile regions such as

high elevation ecosystems of the Mediterranean alpine realm, an environmentally

vulnerable biodiversity "Hot-Spot" (Giorgi, 2006; Schröter et al., 2005) where recent

climate change is affecting species richness and distribution (Médail and Quézel, 1999;

Pauli et al., 2012). Therefore, alpine wetlands in the Mediterranean region, such as the

ones from the Sierra Nevada in the southern Iberian Peninsula, are sensitive recorders of

changing climate and their sedimentological records archive the ecological and

biogeochemical responses to different environmental forcings (Catalan et al., 2013).

In order to contribute to a better understanding of recent climate change events in

these vulnerable areas, here, for the first time, we calibrate a recently developed algal

lipid-derived temperature proxy in an alpine lacustrine record that overlaps with

instrumental temperature time-series. This calibration allows the reconstruction of

temperatures in alpine areas of the southern Iberian Peninsula during the Common Era when instrumental records are discontinuous or non-existent. Temperature-dependent biomarkers, such as those produced by algae (alkenones) or bacteria/archaea (glycerol dialkyl glycerol tetraethers: GDGTs) have been commonly used in a wide range of marine records as quantitative paleothermometers, and their further application in lake environments has widely increased in the last decade (e.g., Castañeda and Schouten, 2011; Colcord et al., 2015; Foster et al., 2016; Longo et al., 2018; Theroux et al., 2010). Another promising type of algal lipid biomarkers, the long-chain alkyl diols (hereafter LCDs), have also been assessed as temperature proxy in marine environments (Rampen et al., 2014b; Rampen et al., 2012; Rodrigo-Gámiz et al., 2014; Rodrigo-Gámiz et al., 2015). Nevertheless, the relationship between LCDs and temperature has only been tentatively tested in freshwater environments (Rampen et al., 2014a). In this regard, studies using LCDs as (paleo)environmental proxies in marine environments (not just only for temperature reconstructions) have increased in the last years, showing the potential of LCDs as proxies for upwelling (Rampen et al., 2008; Versteegh et al., 1997; Willmott et al., 2010), riverine inputs to marine settings (de Bar et al., 2016; Lattaud et al., 2017a; Lattaud et al., 2018a), or nutrient inputs (Gal et al., 2018). Nevertheless, only a few studies have tested LCDs as lacustrine archives of paleoproductivity (Shimokawara et al., 2010), past rainfall anomalies (Romero-Viana et al., 2012), or temperatures (Rampen et al., 2014a), among others. In any case, despite the great potential of LCDs for paleoenvironmental reconstructions, a number of questions exist about the applicability of diols in high latitude areas (Rodrigo-Gámiz et al., 2015), in freshwaters records (Rampen et al., 2014a), and about the distribution and sources of the biological producers (Balzano et al., 2018; Villanueva et al., 2014; Yu et al., 2018).

The LCD distribution in marine environments shows significant correlations with
mean annual sea surface temperature through the ratio of the fractional abundances of $C_{28}$
1,13-diol, $C_{30}$ 1,13-diol, and $C_{30}$ 1,15-diol that are used in the Long chain Diol Index (LDI,
Eq. (1)) (Rampen et al., 2012). The application of LCDs as a temperature proxy is novel
in freshwater environments and only two preliminary calibrations based on recent surface
sediments have been obtained using both mean annual air temperatures (weather station
data) and organic-derived temperature proxies (GDGTs) (Rampen et al., 2014a). Here,
we improve the biomarker paleothermometry by establishing the first temperature
calibration for freshwater LCDs using a comparison with historical temperature records
for the last ~100 years. Although this calibration can only be applied to the studied lake
at present, and perhaps to other alpine wetlands in the Sierra Nevada area, these new data
support and reinforce the promising use of LCDs as a paleotemperature proxy in
freshwater environments.

Equation (1) LDI= ($F_{C_{30}}$ 1,15-diol)/($F_{C_{28}}$ 1,13-diol + $F_{C_{30}}$ 1,13-diol + $F_{C_{30}}$ 1,15-diol)
(Rampen et al., 2012)

**1.1 Regional settings**

This paper focuses on the LCD record of two adjacent cores from Laguna de Río
Seco (LdRS), a small alpine lake (~0.42 ha and less than 3m of water depth) at 3020 masl
in the protected Sierra Nevada National Park, southern Spain (Fig. 1). Alpine Sierra
Nevada wetlands, including LdRS, are low primary production (oligo-mesotrophic)
systems and their biogeochemical cycles partially depends on aeolian nutrient supplies
(e.g., Saharan aerosol deposition), since catchment basins are small and barren in
nutrients (Morales-Baquero et al., 2006; Pulido-Villena et al., 2005; Reche et al., 2009).

Sierra Nevada is the southwestern-most mountain range in Europe, where latest

Pleistocene cirque glaciers carved the metamorphic (mica schist) bedrock in the highest
peaks (Castillo Martín, 2009). Massive glacier melting at the latest Pleistocene-Holocene
transition transformed the former glacial depressions into lacustrine areas (Castillo
Martín, 2009) that evolved gradually into either shallow lakes or peatlands around the
middle-to-late Holocene transition (Garcia-Alix et al., 2017; Jiménez-Espejo et al., 2014).
Small glaciers re-appeared at the highest peaks of the Sierra Nevada in the 15th century,
during the Little Ice Age (LIA), and remained until the 20th century (Oliva et al., 2018).
The presence of these glaciers is observed in the sedimentary record in some alpine lakes
and wetlands in the Sierra Nevada as deposit of coarse sediments, like Laguna de la
Mosca on the north face of the Sierra Nevada (Oliva and Gomez-Ortiz, 2012). However,
these kinds of deposits have not been registered in LdRS (south face of the Sierra
Nevada), where the last 1500 years are characterised by continuous laminated clays and
bryophyte layers (Anderson et al., 2011). Glacial effects have not caused any disturbance
on wetland sedimentation (e.g., erosion) and local alpine sedimentary records show
continuous sedimentation patterns (Anderson et al., 2011; García-Alix et al., 2012;
Jiménez-Moreno and Anderson, 2012; Jiménez-Moreno et al., 2013; Mesa-Fernández et
al., 2018; Oliva and Gomez-Ortiz, 2012; Ramos-Román et al., 2016). Conversely, an
increase in sedimentation rates have been detected in the last ~200 years, probably
resulting from the waning stages of the LIA (Oliva and Gomez-Ortiz, 2012) and enhanced
human activities in the alpines areas of Sierra Nevada during the 19th (García Montoro et
al., 2016; Titos Martínez, 2019; Titos Martínez and Ramos Lafuente, 2016) and 20th

(Jiménez et al., 2015) centuries. These high-sedimentation rates did not affect the natural responses of the local algal communities to environmental variables such as temperature, but there has been a dilution effect of algal compounds (e.g., chlorophylls and labile carotenoids) in the sediments (Jiménez et al., 2015).

During the 20th century this sensitive alpine region of southern Iberia has experienced significant impacts from modern climate change as evidenced, for example, by the first permanent European glacier loss there during the first half of the 20th century (Grunewald and Scheithauer, 2010) and the extreme permafrost reduction during recent decades (Oliva and Gomez-Ortiz, 2012). This melting supplied a large volume of freshwater (Jiménez et al., 2019) that boosted the water availability in the area and the occasional development of local aquatic environments, contrasting with the general environmental aridification trend observed throughout the 20th century (Garcia-Alix et al., 2017; Jiménez et al., 2019; Ramos-Román et al., 2016).

The sedimentary archive of LdRS has been selected for this study in order to 1) improve the freshwater LCD paleothermometry by proposing a new temperature calibration for freshwater LCDs in the alpine wetlands of the Sierra Nevada area, 2) reconstruct temperatures beyond the instrumental record in a site at the leading edge of changing climate, 3) assess the role of different radiative forcing (e.g., solar radiation or greenhouse gas concentrations) on temperature change in alpine wetlands of the southwestern Europe during the Common Era, and 4) understand the responses to recent climate change in this highly sensitive environment.

**2. Materials and methods**

**2.1. Sediment sampling**

Two sediment cores were taken at the deepest part of LdRS, an alpine lake at 3020 masl in the Sierra Nevada (southern Iberian Peninsula) (Fig. 1). A long sediment core (150 cm) was retrieved in 2006 (LdRS lgc). A short sediment core of 16 cm was collected in 2008 (LdRS shc) using a slide-hammer gravity corer (Aquatic Research Instruments, Hope, Idaho, USA). Independent age models were performed in each sediment core to avoid potential correlation problems caused by changes in the sedimentation rates between both coring sites (Fig. 1c) and different sampling dates (2006-LdRS lgc and 2008-LdRS shc). The age model of LdRS lgc is based on $^{210}$Pb and $^{137}$Cs in the uppermost part (first 15 cm), and $^{14}$C analyses in older sediments (Anderson et al., 2011). The age model of the LdRS shc is based on gamma spectroscopy by measuring the $^{210}$Pb, $^{137}$Cs, and $^{226}$Ra radionuclides in the first ~14 cm, and afterwards the age was extrapolated to the core bottom (16 cm) (Jiménez et al., 2019; Jiménez et al., 2018). Both records show that the sediment accumulation rate for the uppermost 15-16 cm ranges between 0.09 and 0.13 cm/yr (Anderson et al., 2011; Jiménez et al., 2018), with lower sedimentation rates below this depth (~0.008 cm/yr) (Anderson et al., 2011). Ages models show that the LdRS shc extends back to ~200 years with a sample resolution ranging from 5 to 7 years (high-resolution) (Jiménez et al., 2019; Jiménez et al., 2018). In the case of the LdRS lgc, the section studied in this paper covers the last ~1500 years with a lower sample resolution. In this case, the sample resolution is around 6-7 years in the first 10 cm, and from 24 to 150 years in older samples (Anderson et al., 2011).

**2.2. Geochemical analyses**

Thirty-two sediment samples were collected consecutively every 0.5 cm along LdRS shc and twenty-one samples in the first 22 cm of LdRS lgc. The samples were freeze-dried and homogenized. The total lipid content was extracted from the sediment samples using a Thermo Scientific™ Dionex™ ASE™ 350 Accelerated Solvent Extractor system at 100ºC and $7 \times 10^6$ Pa using a mixture of dichloromethane (DCM) and methanol (9:1, v:v). Afterwards, the neutral fraction was separated by means of aminopropyl-silica gel chromatography using DCM:isopropanol (1:1, v:v). This neutral fraction was subsequently eluted with hexane, DCM, ethyl acetate:hexane (25:75, v:v), and methanol through a 230-400 mesh/35-70 micron silica-gel chromatographic column, in order to obtain four neutral sub-fractions (N1-N4). Long chain diols were obtained in the third neutral fraction (N3, alcohol fraction), which was derivatised by bis-(trimethylsilyl) trifluoroacetamide (BSTFA) before running the analyses. 30µL of BSTFA and 40µL of pyridine were added to each N3 fraction and heated at 80°C for 2 hours. When vials were at room temperature, a volume between 140 µl and 220 µl of DCM was added to each sample. Firstly, the derivatised N3 fractions were analysed with a Gas Chromatography with Flame-Ionization Detector (GC-FID Shimadzu 2010). An external standard of cholesterol was measured every five samples in order to estimate the appropriate concentration for mass spectrometry analyses. The sample at 19.5 cm depth in the long core was discarded because its concentration was below detection limits. Subsequently, the N3 fractions were measured in a Shimadzu QP2010-Plus Mass Spectrometer interfaced with a Shimadzu 2010 GC using a scan mode between $m/z$ 50 – 650, in order to obtain a general picture of the mass spectrum of the samples and the specific retention times where the $C_{28}$, $C_{30}$, and $C_{32}$ diols eluted. Afterwards, samples where re-analysed on the basis of a Selected-Ion Monitoring mode (SIM), selecting the

characteristic fragment ions of the most important long chain diols, i.e., *m/z* 299, 313,
327, and 341 (Rampen et al., 2012; Versteegh et al., 1997) and the specific retention time
window to identify the $C_{28}$, $C_{30}$, and $C_{32}$ diols with the mid-chain alcohol positioned at
carbon 13, 14 or 15. Fractional abundances of the $C_{28}$ 1,13-diol, $C_{30}$ 1,13-diol, $C_{30}$ 1,15-
diol were used in Eq. (1) to calculate the Long chain Diol Index (LDI) (Rampen et al.,
2012). Fractional abundances of $C_{28}$ 1,13-diol, $C_{30}$ 1,13-diol, $C_{30}$ 1,15-diol and $C_{32}$ 1,15-
diol were used to characterise the potential diol source (e.g., marine, lacustrine, or specific
algae groups) (Lattaud et al., 2018a; Rampen et al., 2014a). Fractional abundances of the
$C_{28}$ and $C_{30}$ 1,14-diols were measured only in the short core in order to assess their
potential relationship with temperatures (Rampen et al., 2014b). The presence of the $C_{32:1}$
1,15-diol has also been tested, but it was only identified in some samples from the short
core at very low concentrations, thus is not included it in this study.

**2.3. Reference temperature time-series for LCD temperature calibrations**

Generating an accurate temperature calibration based on LCDs in alpine wetlands

from the Sierra Nevada area is challenging, because there is a lack of long and continuous
temperature time-series at such high elevations. The meteorological observatories at the
Sierra Nevada ski resort (ranging in elevation from 2500 to 3020 masl) only provided
discontinuous temperature records from 1965 to 2011 (Observatorio del cambio global
de Sierra Nevada, 2016; Spanish National Weather Agency - AEMet Open Data, 2019)
that show a significant correlation (r>0.95; p<0.0001) with low elevation temperature
time-series (Table S1, S2). Therefore, a potential way to obtain a LCD-based temperature
calibration is by means of the correlation of LCD data with long and reliable historical
temperature time-series at nearby lower elevation areas, followed by a correction for the
altitudinal effect on temperatures.

The three weather observatories in the Granada area, at the foothills of the Sierra
Nevada, only provide reliable temperature data from the 1970s onwards (Spanish
National Weather Agency - AEMet Open Data, 2019), which is a short-period for an
accurate down-core proxy calibration. Temperature time-series preceding the 1970s have
been reconstructed using statistical models (e.g., Gonzalez-Hidalgo et al., 2015), and
show a good correlation with the LDI and with the relative abundance of the $C_{28}$ 1,13-,
$C_{30}$ 1,13-, and $C_{32}$ 1,15-diols, with a weaker correlation for the $C_{30}$ 1,15-diol (Table S3).
However, these correlations are weaker than the ones obtained from Sevilla-Tablada and
Madrid-Retiro observatories. These observatories registered longer and more reliable
temperature data than those obtained in Granada observatories, which are likely biased
by the quality of the reconstructed temperature data. Therefore, after testing the
correlations between LCDs and different low elevation observatories (Table S3), we
decided to develop the LCD-based temperature calibrations against the temperature time-
series from Sevilla-Tablada and Madrid-Retiro observatories. These observatories show
the best correlations with the LCD data, in addition to being the most reliable and longest
temperature time-series in the region (see Table S3 for further explanations and Fig. 1a
for the location of these low-elevation observatories).

Another question to clarify in the study records before selecting the reference
temperature time-series for the LCD calibration is the potential seasonal effect on the
LCD distributions, since different studies have shown diverse relationships between
annual or seasonal temperatures and the LCDs. For example, good correlations have been
found in marine environments between the fractional abundances of LCDs (expressed as
LDI in all the cases) and annual (Rampen et al., 2012), winter and annual (Smith et al.,
2013), or autumn and annual sea surface temperatures (Lattaud et al., 2018b). Despite
fewer studies on the LCD distribution in freshwater environments, Rampen et al. (2014a)
found a good correlation between the LCD distributions in a suite of lake surface
sediments and mean summer lake temperatures (deduced from GDGTs). Nevertheless,
the direct correlation between these LCD distributions and annual or seasonal air
temperatures was weaker, probably due to the location of the weather observatories with
respect to each study area. Villanueva et al. (2014) also investigated this seasonal effect
and detected changes in the LCD distribution throughout the year in the water column
and surface sediments from an African lake that could be either related to successive and
different LCD-producer blooms or seasonal variations in the LCD production by a unique
source. Both scenarios might affect LCD-based temperature reconstructions.

Considering all these constrains to select the best temperature time-series to

establish an accurate LCD-based temperature calibration, the most rigorous approach for
the studied alpine site would consider annual and monthly water and/or air temperatures
of the catchment basin at 3020 masl, as well as, the periods of the year when the LCDs
are produced, but these data are not available so far. Thus, the effect of seasonality has
been estimated by means of the comparison with reliable seasonal long temperature time-
series from lower elevation sites. In this regard, seasonal air temperatures for the last ~100
years registered in Madrid and Sevilla observatories correlate with the LCD distributions
(with a weaker correlation for the $C_{30}$ 1,15-diols) and the LDI ($0.9 > r > 0.6$; $p < 0.001$).
Nevertheless, this correlation is generally lower than the one obtained when considering
only mean annual air temperatures (MAAT) (i.e., in the case of the LDI $vs$ MAAT $r=0.9$;
$p<0.0001$). Since warm temperatures influence the algae growth in the studied area
(Carrillo et al., 1991; Sánchez-Castillo, 1988), we would expect a higher correlation
between LCDs and mean seasonal air temperatures from the warmer months (MWAT:
May-September), which is potentially the LCD production season, but this correlation is
lower than the annual ones in the case of the LDI (LDI *vs* MWAT $0.8>r>0.7$; $p<0.0001$)
(Table S3, S4). A similar pattern is observed when annual and warm season temperatures
are compared with the fractional abundances of the $C_{28}$ 1,13-, $C_{30}$ 1,13-, $C_{30}$ 1,15-, and
$C_{32}$ 1,15-diols (Table S3). Consequently, in view of 1) the fact that this is the first attempt
at a freshwater LCD-based temperature calibration in this area; 2) there exists a high
correlation between the different instrumental time-series of regional air temperatures
(seasonal *vs* annual), and 3) the best correlations (normal and detrended) between the
LCD distributions and temperatures are obtained when using MAAT (Tables S3, S4), we
use MAAT for the LCD-based temperature calibrations in this study. Nevertheless,
further work, including a monitoring program for monthly air temperatures in the
catchment area and water temperatures in the lake, as well as, suspended particulate
matter and sediment trap studies, is required to better understand the local LCD
production, improve the LCD-based temperature calibration, and minimise the
uncertainties of the current approach.

Two groups of reference temperature time-series at 3020 masl, based on the same

batch of data, have been estimated in order to overcome the scarcity of high-elevation
temperature time-series in the Sierra Nevada and obtain a reliable mean LCD-based
temperature calibration: 1) based on the elevational gradient between low and high
elevation observatories and 2) based on the direct correlation between temperature time-
series from Madrid and Sevilla observatories and that at 3020 masl (Cetursa 5
observatory) in the Sierra Nevada, which is near LdRS and at the same elevation (Table
S1).

Reference temperature time-series 1: The environmental lapse rate

($\Delta_{temperature}/\Delta_{elevation}$ in ºC/m) between lower elevation observatories (with long
temperature time-series: Granada, Sevilla, and Madrid) and those from Sierra Nevada at
higher elevation (with shorter temperature time-series: Albergue, and Cetursa 1, 3 and 5)
has been estimated in order to correct the elevational gradient between them (more than
2200 m: Table S1). Due to few annual data points from high elevation sites, monthly and
annual (twelve continuous months) environmental lapse rates were calculated to compare
both datasets. The calculated temperature shifts between the reference low elevation
observatories and LdRS site at 3020 masl, worked out from Fig. S1 equations (Table S5),
was applied to the temperature time-series from Madrid and Sevilla for the last ~100 years
in order to obtain two reference temperature reconstructions (from 1908 to 2008 CE) at
3020 masl: reference temperature time-series 1a (from Madrid data), and reference
temperature time-series 1b (from Sevilla data).

Reference temperature time-series 2: The direct comparison between Madrid and

Sevilla temperatures and those from the observatory Cetursa 5 (3020 masl) by means of
ordinary least square regressions has given rise to two equations (Fig. S2a and b) that
allow the reconstruction of temperature time-series at 3020 masl from 1908 to 2008:
reference temperature time-series 2a (from Madrid data), and reference temperature time-
series 2b (from Sevilla data).

As a result, we have obtained four reference temperature time-series at 3020 masl
where the effect of the altitudinal difference between low elevation observatories (Madrid
and Sevilla) and LdRS have been corrected by two different methods. Consequently,
these four reference temperature series are highly similar, showing a certainly high
correlation ($r>0.98$; $p<0.0001$), without significant difference between the sample
medians (deduced from a Kruskal-Wallis test), and very low standard deviation between
samples from the same time interval (SD 0.2).

**3. Results**

**3.1. Long chain diols in the LdRS records**

Six main LCD isomers have been identified and relative abundance analysed in
the LdRS cores: $C_{28}$ and $C_{30}$ 1,13-diols, $C_{28}$ and $C_{30}$ 1,14-diols, and $C_{30}$ and $C_{32}$ 1,15-
diols. The LCD abundance changes through time in both records, but the $C_{32}$ 1,15-diol is
the predominant isomer in most of the samples. Nevertheless, the relative abundance of
the $C_{32}$ 1,15-diol drops abruptly (relative abundances between 25% and 40%) during the
LIA, contrasting with a relative increase in the $C_{28}$ and $C_{30}$ 1,13-diols (Fig. 2). This switch
in the most abundant isomers can be read as either a change in the LCD producers or an
adaptation to colder temperatures of the same organism, and thus affecting the LCD
production. Conversely, the $C_{28}$ and $C_{30}$ 1,14-diols show the lowest relative abundances
($1.3 \pm 0.4$% and $1.8 \pm 0.3$%, respectively), and were only quantified in the short core to
assess their potential application as paleothermometer in LdRS. Although they show a
good correlation with temperatures ($C_{28}$ 1,14-diol: $0.66>r>0.45$ $p<0.04$; $C_{30}$ 1,14-diol: -
$0.82<r<-0.53$ $p<0.02$), their low relative abundance, very close to the detection limit,
together with a different biological source (Sinninghe Damsté et al., 2003), preclude us
from including them in the temperature calibration, and therefore, in the discussion of
this study. Consequently, the interpretations in this paper are only focused on the
distribution pattern of the relative abundances of the main LCDs in LdRS: $C_{28}$-$C_{30}$ 1,13-
diols and $C_{30}$-$C_{32}$ 1,15-diols.

The relative abundances of these four isomers in LdRS correlate well in both short

and long cores, only $C_{30}$ 1,15-diol showing weaker correlations with the other isomers
(Table S6). Overall, $C_{28}$ and $C_{30}$ 1,13-diols show opposite trends to those from the $C_{30}$
and $C_{32}$ 1,15-diols. Their general trends for the last ~100 years seem to be influenced by
the temperature oscillations at 3020 masl (Table S7): the $C_{28}$ and $C_{30}$ 1,13-diols display a
negative correlation with temperatures ($r$<-0.7 $p$<0.0001) and the $C_{32}$ 1,15-diol a positive
one ($r$>0.8 $p$>0.0001). Although the $C_{30}$ 1,15-diol also shows a positive relationship with
temperatures, this correlation is weak ($r$>0.3 $p$>0.0001). Accordingly, the LDI values
from LdRS for the last ~100 years show a significant correlation ($r$>0.9 $p$>0.0001) with
the reference temperature time-series at 3020 masl (Table S7).

The different diol isomers in LdRS also show good agreement with general

temperature trends for southwestern Europe during the last ~1500 years (Abrantes et al.,
2005; Luterbacher et al., 2016; Nieto-Moreno et al., 2013; Sicre et al., 2016, among
others). The $C_{30}$ and $C_{32}$ 1,15-diols depict a positive relationship with temperatures,
whereas the $C_{28}$ and $C_{30}$ 1,13-diols display a negative one. Thus, the LDI record obtained
from the $C_{28}$-$C_{30}$ 1,13-diols, and $C_{30}$ 1,15-diol also show important fluctuations during
the last ~1500 years, in agreement with the general temperature trends of the Common
Era (CE). More specifically, LDI values in the LdRS lgc range from ~0.23 to 0.05 from
~400 to 1900 CE, with maximum and minimum values recorded at ~930 and ~1690 CE,
respectively (Fig. 3). These changes are coeval with the minimum temperatures of the
LIA and the maximum temperatures of the Medieval Climate Anomaly (MCA) in Europe
(e.g., Luterbacher et al., 2016; Nieto-Moreno et al., 2013; Sicre et al., 2016). The rates
of change were higher during the $20^{th}$ century, with LDI values ranging from 0.10 to 0.31
in the lowest resolution LdRS lgc record and from 0.13 to 0.32 in the highest resolution
LdRS shc record. These minimum and maximum values were reached in both cases
during the first and last decades of the $20^{th}$ century, respectively (Fig. 3).

**3.2. LCD temperature calibration**

A total of twenty-six samples from both short and long LdRS cores ranging in age

from 1908 to 2008 were selected to perform the LCD-based temperature calibration,
along with the two groups of reference temperature time-series at 3020 masl. Since
sedimentary samples used in the calibration have a time averaging between 5 and 7 years,
a mean of the historical temperatures covering the same time averaging of each sample
was calculated.

Eight different calibrations have been performed: five using the LDI, one using a

multiple linear regression of the relative abundances of $C_{28}$ 1,13-diol, $C_{30}$ 1,13-diol, and
$C_{30}$ 1,15-diol (following Rampen et al., 2014a) (MLR calibration 1 hereafter), one using
multiple linear regressions of the ratios of the relative abundances of LCDs with positive
(even weak) correlation with temperature against the ones with negative correlation ($C_{30}$
1,15- / $C_{28}$ 1,13-diols; $C_{30}$ 1,15- / $C_{30}$ 1,13-diols; $C_{32}$ 1,15- / $C_{28}$ 1,13-diols; and $C_{32}$ 1,15-
/ $C_{30}$ 1,13-diols) (MLR calibration 2 hereafter), and one using multiple linear regressions
of the ratios of the relative abundances of $C_{30}$ 1,15- / $C_{28}$ 1,13-diols and $C_{30}$ 1,15- / $C_{30}$
1,13-diols (MLR calibration 3 hereafter). The statistics and the equations for the MLR
calibrations 1, 2, and 3 are described in the Table S8.

In the case of the LDI, ordinary least square regressions were run between the four

reference temperature time-series at 3020 masl and the LDI record from LdRS shc and
lgc, resulting in four calibration equations (Fig. S3). The slopes of these four equations
range from 8.2 to 10.2. The LDI-derived temperatures from the reference time-series 2
show the highest values for the last ~100 years, whereas the minimum values are mainly
shown by the ones calculated with the reference time-series 1. The difference between
the four LDI-derived temperatures for the last ~100 years is low, with a standard deviation
lower than 0.13. The standard error of these four individual calibrations ranges from 0.18
to 0.23ºC, and the maximum residual is ~0.8ºC. However, due to the uncertainty of
establishing an accurate temperature time-series at 3020 masl, LDI-derived temperature
values from these LDI individual calibrations have been used to determine the range of
the variation (minimum and maximum temperature values) for each point, and an
additional calibration, summarising the relationship between LDI and the four reference
temperatures at 3020 masl has been performed. The obtained 104 combinations of LDI
and temperature data provided an equation representing the average relationship between
MAAT and LDI (Eq. (2); Fig. 4a). Since this is a summary of the four temperature time-
series, the residual errors include the residual errors of the individual LDI calibrations
(Fig. S3). The residual errors of this average temperature calibration, according to both
the LDI-reconstructed temperatures and the reference temperature time-series, are lower
than 0.8ºC (similar to the four individual LDI calibrations), with the standard error of
0.28ºC. The histogram showing the frequency of the residuals reveals that the ~85% of
the residuals range from 0.4 to -0.4ºC. This percentage is slightly lower (~62%) when the
residual interval is established between 0.2 and -0.2ºC (Fig. 4b). Only one data point from
1973 among the 104 data combination may be an outlier since it shows a residual 2.5
times higher than the residual standard deviation.

Equation (2) MAAT (ºC) = 9.147 x LDI - 0.243          (n = 26 x 4; $r^2$ = 0.79)*
* n = 26 LDI values plotted against four reference temperature time-series providing a
total of 104 combinations.

All the calibrations (LDI and MLR calibration 1, 2, and 3) show good correlation

with temperatures (Fig. 4 for LDI and Table S8 for MRL calibrations 1, 2, and 3). The
obtained temperatures from the average LDI calibration and those using multiple linear
regressions depict very similar trends in both cores (Fig. S4; $r$>0.96; $p$<0.0001).
Nevertheless, the correlation is slightly lower ($r$>0.82; $p$<0.0001) for the results from
MLR calibration 1 in the long core. In addition, some inconsistencies come up in the
reconstructed temperatures from MLR calibration 1: in general, temperatures tend to be
lower than those from the other calibrations, giving rise to negative annual values during
the LIA. This would be highly unlikely since under this scenario the lake would have
stayed frozen all year round, with little or no sedimentation occurring. Temperature
reconstructions using MLR calibrations 2 and 3 take the advantage of the ratios of isomers
with positive $vs$ negative relationship with temperatures. The results are similar to the
ones obtained with the LDI, except for the LIA. Nevertheless, we discarded MRL
calibration 2 as it also uses the $C_{32}$ 1,15-diol, whose relationship with temperature is not
clear, only reporting a positive correlation in some culture studies (Rampen et al., 2014b).
MRL calibration 3 provides the most similar results to those from the LDI, especially
comparing the temperature anomalies of both records with respect to the last 30 years of
the record, showing differences of less than ~0.1ºC. However, this difference increases
(~0.3ºC) in the LIA. As this is the first study of LDI in sedimentary records from alpine
lakes of the Sierra Nevada area, we have opted for a conservative solution following the
LDI temperature calibration. The temperature reconstructions from MRL calibration 1,
2, and 3 will only be mentioned when needed in the discussion.

The application of the obtained calibration to the LDI values of LdRS (Eq. 2)

produced the first temperature reconstruction for the Common Era in this alpine area (Fig.
3). Nevertheless, a potential challenge of using this kind of down-core proxy calibrations,
is that the uncertainty of the reconstructed variables (temperature in this case) would
increase when data fall outside the calibration data-set (e.g., during the LIA). Further
studies on the local LCD production in this alpine area will contribute to extend the range
of temperatures in the calibration, reducing the uncertainties of the LCD-derived
temperatures.

In order to estimate the real magnitude of temperature variations during the

Common Era, the mean annual air temperature anomaly (MAATA ºC) has been
calculated with reference to the annual MAAT of the last 30 years of the record (2008-
1979). The lowest temperatures were recorded between ~1600 and ~1780 CE, with a
temperature anomaly ranging from ~ -1.9 to ~ -2.2ºC. These temperature anomalies only
reached positive values after 1998 (Fig. 3).

**4. Discussion**

**4.1. Long chain diol distribution in alpine lakes from southern Iberia**

The distribution pattern of the main LCDs in LdRS ($C_{28}$, $C_{30}$ 1,13- and $C_{30}$, $C_{32}$ 1,15-diols) could help us decipher the potential biological producers. There is a high percentage of the $C_{32}$ 1,15-diol, which is one of the main features observed in freshwater environments (Lattaud et al., 2018a; Rampen et al., 2014a) (Fig. 2, S5). Another relevant feature of the distribution of the LCDs in LdRS is the low and constant relative abundance of the $C_{30}$ 1,15-diol (6.5 ± 1.5%) throughout the LdRS records (Fig. 2), agreeing with the range of the most probable distribution of $C_{30}$ 1,15-diol in marine algae (Fig. S5; Table S9). This feature is not common in marine or lake sediments (de Bar et al., 2016; Lattaud et al., 2017a; Rampen et al., 2014a), resulting in an almost unique area for LdRS isomers (most specifically $C_{28}$ 1,13-, $C_{30}$ 1,15-, and $C_{32}$ 1,15-diols) when comparing with literature data in a ternary diagram (Fig. 2). Conversely, the distribution of both $C_{28}$ and $C_{30}$ 1,13-diols usually shows similar patterns as other freshwater samples (Lattaud et al., 2018a; Rampen et al., 2014a) (Fig. 2, S5, Table S9). A Kruskal-Wallis ANOVA test was used to assess whether the distribution of the main LCDs in LdRS and from other sources (e.g., marine, freshwater, algal culture) were statistically different. Results point towards no significant differences between the whole LCD distribution in LdRS and the other sources. Nevertheless, the Kruskal-Wallis test found significant differences among individual isomers of the different sources (including LdRS). Subsequently, a Mann-Whitney U test was performed to compare pair of groups (individual isomers from LdRS *vs* individual isomers from source 1, 2, and so on), finding significant differences among most of them (Table S9). All these evidences suggest that the LCD distribution in the LdRS might differ from those of previous studies published to date and the potential biological producers at LdRS would be thus uncertain.


Eustigmatophyceae algae (e.i. *Vischeria* sp., *Eustigmatos* sp.) have been
commonly proposed as the main diol producers in freshwater environments dominated
by a mix of $C_{28}$ 1,13-, $C_{30}$ 1,15-, and $C_{32}$ 1,15-diols (Rampen et al., 2014a; Villanueva et
al., 2014; Volkman et al., 1999). Moreover, a dominance of $C_{32}$ 1,15-diol has been
identified in families of Goniochloridaceae and Monodopsidaceae (Lattaud et al., 2018a;
Rampen et al., 2014a). Nevertheless, planktonic algae communities are very simple in the
alpine Sierra Nevada wetlands (Sánchez-Castillo, 1988) and Eustigmatophyceae algae
have not been identified so far (Barea-Arco et al., 2001; Sánchez-Castillo, 1988). Thus,
this study suggests that LCD producers in LdRS might be different from those identified
previously in other freshwater environments and algal culture studies, making the
potential source of LCDs even more complex than originally thought. Consequently, the
outcomes of this paper (i.e., the LCD-based temperature calibration) should not be
generally applied to other freshwater records unless they show a similar LCD distribution
as LdRS. Additional research combining lipids and 18S rRNA gene sequencing analyses
from suspended particulate matter, surface sediments, and sediment traps would be
needed to unravel the real biological sources of LCDs in these alpine wetlands.

**4.2. LdRS record in the environmental context of the Iberian Peninsula during the**
**Common Era**

Abrupt changes in temperature and precipitation have been depicted during the
last 2000 years in the Iberian Peninsula and surrounding marine areas (Moreno et al.,
2012; Sánchez-López et al., 2016). Precipitation was highly variable, showing arid
conditions during the MCA, especially in southern Iberia, overall humid conditions
throughout the LIA (with a complex internal structure showing large variability in
humidity and extreme events), and arid conditions for the Industrial Period (Moreno et
al., 2012; Oliva et al., 2018; Rodrigo et al., 1999; Sánchez-López et al., 2016), especially
in high elevation wetlands from southern Iberia (Anderson et al., 2011; Garcia-Alix et al.,
2017; Jiménez-Espejo et al., 2014).

Although the Early Middle Ages displayed a great temperature variability in the

Iberian Peninsula and surrounding marine sites (Moreno et al., 2012; Sánchez-López et
al., 2016), three main stages have been identified for the last millennium deduced from
different proxies: a warm period throughout the MCA followed by cold temperatures
during the LIA, ending in an abrupt warming in the second half of the 20$^{th}$ century
(Moreno et al., 2012; Oliva et al., 2018; Sánchez-López et al., 2016). One of the proxies
used to reconstruct such temperature variations in continental areas of the Iberian
Peninsula has been tree ring data. Long tree ring temperature archives of the Iberian
Peninsula showed the same overall variations as the ones registered in LdRS, such as high
temperatures before 1250 CE (Büntgen et al., 2017), some temperature declines coeval
with solar minima during the LIA (e.g., the end of Spörer or Maunder Minima), as well
as a period of moderate-low temperatures from ~1850 to ~1940, followed by an
increasing temperature trend in the second half of the 20$^{th}$ century with several
temperature drops between ~1960 and ~1990 (Büntgen et al., 2017; Tejedor et al., 2017).
Nevertheless, the warming documented from the LCD-derived temperatures in the last
stages of the LIA is more pronounced in the LdRS record. The same overall trends have
been observed in European summer temperatures deduced from tree ring records
(Luterbacher et al., 2016) (Fig. 5b, d, and 6b, c). Surprisingly, tree ring data from the
Pyrenees and other Iberian areas show minor temperature variations, and even a slight
temperature decrease from ~2000 to 2008 similar to the one observed in the LCD-derived
temperatures from the LdRS record (Fig. 4c). This temperature stabilization at the
beginning of the 21$^{st}$ century is coeval with globally reduced warming rates over the
2001–2014 period (Fyfe et al., 2016).

Contrasting with these continental temperature reconstructions, high-resolution
sea surface temperature (SST) estimations from marine sites surrounding the Iberian
Peninsula, such as those derived from alkenones ($U^{K'}_{37}$) in the Tagus Delta (Iberian
Atlantic Margin) or in the Balearic Islands (western Mediterranean Sea), showed a
general decreasing trend for the last ~2000 years, with a warm MCA, a cold LIA, and
cold/moderate temperatures for the Industrial Period that do not appear to mirror the
modern global warming observed throughout the 20$^{th}$ century (Abrantes et al., 2005;
Moreno et al., 2012). Only high-resolution $U^{K'}_{37}$-and $TEX_{86}$-derived (from GDGTs) SST
records of the cores 384B and 436B from the Alboran Sea (Nieto-Moreno et al., 2013)
and the $U^{K'}_{37}$-SST record of core Gol-Ho1B from the Gulf of Lion (Sicre et al., 2016)
have shown a clear temperature increase during the 20$^{th}$ century, similar to the LDI
temperature record in LdRS (Fig. 5a,d, 6a,c). The observed heterogeneity in the SST
reconstructions based on different biomarkers such as alkenones (Abrantes et al., 2005;
Moreno et al., 2012; Rodrigo-Gámiz et al., 2014), GDGTs (Nieto-Moreno et al., 2013) or
LCDs (Rodrigo-Gámiz et al., 2014) could be explained since each record belongs to a
different biogeographical area influenced by specific temporal and dynamic
oceanographic regimes, as well as by different primary productivity patterns of each
biological source (e.g., seasonality or bloom length) (Sicre et al., 2016).

The previously described climate variability in precipitation and temperature
during the last ~1500 years in the Iberian Peninsula have been explained by different
forcing mechanisms such as the effect of the westerlies-North Atlantic climate dynamics,
internal climate variability, solar irradiance, volcanism, or anthropogenic forcing
(Gómez-Navarro et al., 2011; Gómez-Navarro et al., 2012; Moreno et al., 2012; Sánchez-
López et al., 2016). Their potential effect on the LCD distribution in the LdRS records is
discussed in the following section.

**4.3. Control mechanisms on alpine temperatures in SW Europe during the**
**Common Era**

This discussion is based on the LDI-temperature reconstruction. However, similar
results are obtained when comparing the temperature reconstructions from MLR
calibrations 1, 2, and 3 in LdRS and the different forcing mechanisms assessed in this
section (Tables S10-S13).

Solar, volcanic, and anthropogenic (e.g., $CO_2$ and $CH_4$) radiative changes, along
with the internal variability are usually attributed as the leading factors controlling
temperatures during the Common Era (Ammann et al., 2007; IPCC, 2013). In addition,
North Atlantic climate dynamics such as the North Atlantic Oscillation (NAO) or the
Atlantic Multidecadal Oscillation (AMO) are other potential drivers of natural climate
variability in the Iberian Peninsula (López-Moreno et al., 2011; Moreno et al., 2012;
O'Reilly et al., 2017; Sánchez-López et al., 2016). The control of the North Atlantic
climate dynamics in the studied alpine wetlands is evident, at least for precipitation and
humidity fluctuations, since the NAO and solar forcing have been described as the main
controls on the paleoenvironmental evolution recorded in this area (Garcia-Alix et al.,
2017; Ramos-Román et al., 2016). Conversely, other studies have shown that the NAO
climate mode had little effect on temperatures in this alpine area from 1950 to 2006 CE
(López-Moreno et al., 2011). LdRS data agree with this observation, since no correlation
(Tables S10-S13) has been detected between the NAO reconstruction (Trouet et al., 2009)
and the obtained LCD record for the last millennium. The AMO has an impact on the
North Atlantic atmospheric blocking mechanisms (Häkkinen et al., 2011) and on the
European and Mediterranean temperatures, especially during the AMO warm phases
(O'Reilly et al., 2017). In the study area, the AMO shows a moderate long-term
correlation (Figs. 5d,e and 6c,d $r>0.60$; $p<0.01$) with both long and short core derived-
LDI records, but the correlation decreases when long-term trends are removed ($r<0.32$;
$p>0.1$) (Tables S10, S11). Since the nature of the AMO and its specific drivers are still
a matter of debate, i.e., internal ocean variability control (multidecadal fluctuations in
the Atlantic Meridional Overturning Circulation) versus solar or volcanic forcing for
the last centuries (Knudsen et al., 2014), we cannot conclude whether the observed
correlations represent the sole effect of the AMO or the influence of its underlying
forcing mechanisms.

The significant correlation at long and short terms ($r>0.61$; $p<0.005$) between

LDI-derived temperatures from LdRS records and greenhouse gases (Schmidt et al.,
2011) (Fig. 6c, g; Table S10), especially since the beginning of the 20th century (Industrial
Period) (Fig. 5d,g; Table S11), suggests that greenhouse gases might have an important
effect on temperatures at this high elevation site.

The potential impact of solar radiation and volcanic eruptions on climate over both
short- and long-time scales is a topic of controversy in the literature (Ammann et al.,
2007). In this regard, volcanic forcing, which should give rise to negative radiative
forcing in the climate system (Ammann et al., 2007; Sigl et al., 2015), do not show a
significant correlation with LDI-derived temperatures from LdRS records over the last
1500 years (Figs. 5d,h and 6c,h; Table S10, S11). We suggest that this lack of influence
at LdRS records is a function of its high-altitude location, at 3020 masl, in the free
troposphere, which reduces the environmental impact of small volcanic tropospheric
eruptions that likely have greater effects on lower elevation sites (Mather et al., 2013). In
addition, the relatively short residence time of volcanic aerosols in the atmosphere mainly
causes, at most, decadal-timescale effects (Sigl et al., 2015) that can be difficult to identify
in most sedimentary records due to the age resolution, as in the case of older sediments
than 200 years in LdRS. Nevertheless, large explosive volcanic eruptions delivering large
amounts of stratospheric aerosols (Marotzke and Forster, 2015; Sigl et al., 2015), such as
that for Agung Volcano in Bali, Indonesia (1963-1964 CE), may be associated with a
small depression in LDI-derived temperatures observed in LdRS records (Fig. 5d,h).
Although cold LDI-reconstructed temperatures occasionally seem to occur coevally with
volcanic eruptions, for example, 560-510 and 320 years ago (~1450-1500 and 1690 CE)
(Sigl et al., 2015), there is no consistent relationship between the intensity - number of
large eruptions and the reconstructed coolings in LdRS records, especially over the last
~200 years when the sample resolution would be enough to detect them (LdRS shc).

Most of the above-mentioned cooling events recorded in LdRS, such as those
during the LIA, are coeval with low solar activity periods like the Spörer Minimum (from
~1450 to 1550 CE) or the Maunder Minimum (from ~1645 to 1715 CE) (Stuiver and
Quay, 1980) (Fig. 6). Thus, long-term correlations between LDI-derived temperatures
and solar activity, based on reconstructions of the solar irradiance and cosmogenic
isotopes (such as $^{14}$C), are evident during the last ~1500 years in LdRS record ($r$>0.69; $p$
<0.002) (Fig. 6c,e,f; Table S11). This correlation drops (0.37< $r$ <0.56 and 0.04< $p$ <0.14)
when long-term trends are removed (Table S11). The long-term solar influence agrees
with previous observations in other alpine records of this area (Garcia-Alix et al., 2017;
Ramos-Román et al., 2016). Solar activity slightly decreases its long-term influence in
LdRS record during the last ~200 years ($r$>0.56; $p$<0.001) and disappears when long-term
trends are removed (Table S10). Only some occasional temperature decreases or slower
rates of warming such as during the 19$^{th}$ to 20$^{th}$ century transition, from ~1930 to 1940,
from ~1960 to 1975, and around 1988 CE, are coeval with slight declines in the total solar
activity (Fig. 5d,f: blue arrows).

In the same way, LdRS registered a small decrease in LDI-derived temperatures

(or stabilization) at the beginning of the 21$^{st}$ century (Fig. 5d), also recorded in the Madrid
and Sevilla temperature time-series (Spanish National Weather Agency - AEMet Open
Data, 2019), and thus in the reconstructed reference temperatures time-series at 3020 masl
(Fig. 5c), in tree ring records of the the Pyrenees and Iberian Range (Büntgen et al., 2017;
Tejedor et al., 2017), in marine platforms of the western Mediterranean (Fig. 5a) (Sicre
et al., 2016), and globally (Fyfe et al., 2016). Although this slowdown agrees with a
decreasing trend in solar activity and a slight stabilization of atmospheric methane
concentrations (Fig. 5f,g), the causes are more complex, and probably related to a
combination of internal variability and radiative forcing (e.g., volcanic and solar
activity, or decadal timescale changes in anthropogenic aerosols) (Fyfe et al., 2016).

**4.4. Exceeding natural thresholds in alpine areas**

The LDI-derived temperatures from LdRS exceeded the highest pre-industrial temperatures in the early 1950s (Fig. 6c), under full anthropogenic influence. The comparison between pre-industrial and post-industrial scenarios in the study site highlights the human impact on natural trends. In this regard, the temperature increase during the last stages of the LIA (from ~1690 to ~1850 CE), an analogue for a non-anthropogenic temperature-increase scenario, was between ~1.2 and ~1.4ºC (~0.09ºC/decade; Fig. 6), whereas the mean temperature rise throughout the 20th century was ~1.8ºC (~0.18ºC/decade; Fig. 7). Both warming rates are roughly similar to those reconstructed from MLR calibrations 1, 2, and 3: ~0.06-0.09ºC/decade for the last stages of the LIA, and ~0.17-0.18ºC/decade for the 20th century. Although this means that on average, the warming rate was two times faster throughout the 20th century than at the end of the LIA (Fig. 7), these observations are based on a low sample density for the LIA (8 samples), which might slightly increase the uncertainty for this period. By comparison, average global temperatures rose by ~0.85ºC from 1880 to 2012 CE, corresponding to 0.06ºC/decade (IPCC, 2013), which highlight the high-elevation amplification effect of temperatures on this vulnerable area.

Other European alpine areas in the Mediterranean region, such as those from the Alps, experienced a slower warming rate during the 20th century (~0.11ºC/decade) (Fig. 7) (Auer et al., 2007; Böhm et al., 2010). This is ~1.6 times slower than the warming rate recorded in the Sierra Nevada. This evidence, along with the generally smaller amount of precipitation in the alpine areas of the western Mediterranean region (Auer et al., 2007;

Rodrigo et al., 1999), allows us to conclude that the 20[th] century environmental stress in
this area was greater than in the Alps.

Future scenarios are not optimistic for Sierra Nevada since temperatures at ~1000

masl may rise between 2.2ºC and 5.3ºC by the end of the 21[st] century (Pérez-Luque et al.,
2016), exceeding the global projections of the IPCC-2013 report (IPCC, 2013). However,
temperature projection and its subsequent impact on alpines areas of the Sierra Nevada
have not been satisfactorily assessed so far due to the lack of long-term quantitative
climatic records at these elevations (e.g., temperature). The LCD-based temperatures at
~3000 masl will solve this lack of quantitative data and will be valuable to project future
scenarios in these alpine ecosystems where endemic and endangered species inhabit
(Blanca, 2001; Munguira and Martin, 1993).

**4.5. Impact on the southwesternmost European alpine glaciers**

The studied alpine area supported the southernmost glaciers in Europe during the

LIA. Glaciers and permanent snow fields below ~3000 masl, such as those of Corral del
Mulhacen (~2950 masl) whose last mention in the literature was between 1809 and 1849
CE (Oliva and Gomez-Ortiz, 2012), would have totally disappeared by the decrease in
regional precipitation during the first half of the 19[th] century (Rodrigo et al., 1999). The
climatic features at the end of the 19[th] century and the beginning of the 20[th] century did
not allow this glacier to re-establish itself (Fig. 8). Post-LIA climatic conditions have also
been proposed as the trigger for the melting of the Corral del Veleta Glacier in Sierra
Nevada (~3100 masl) at the beginning of the 20[th] century (Garcia-Alix et al., 2017; Oliva
and Gomez-Ortiz, 2012; Oliva et al., 2018). However, the LCDs in LdRS records show
that reconstructed temperatures did not exceed the levels of the 1850s until the late 1940s
CE. Precipitation was low in the southern Iberian Peninsula during the first half of the
20th century, but similar, and even lower, precipitation values were registered before
~1850 CE (Rodrigo et al., 1999; Spanish National Weather Agency - AEMet Open Data,
2019) (Fig. 8). Therefore, how could the glacier have retreated under this almost steady-
state scenario? A similar paradox has been described in the Alps (Painter et al., 2013),
where glaciers began to sharply retreat after the mid-19th century, even though
temperature and precipitation records would suggest that glacier expansion should have
occurred at least until the first decades of the 20th century. In this case, one of the proposed
triggers for the glacier retreat was the industrial black carbon deposition that amplified
the solar radiation absorbed at the snow surface and caused its subsequent melting - not
a temperature or precipitation change (Painter et al., 2013). Precipitation data from the
southern Iberia (Rodrigo et al., 1999) along with the LCD-reconstructed temperatures in
LdRS records suggest that temperature and precipitation were not the only drivers of
glacial retreat that led to the melting of permanent glaciers in the Sierra Nevada in the
1920s. Instead, mirroring the case of the Alps, it is plausible that other factors reducing
the albedo, such as enhanced atmospheric deposition may have played a strong role. In
this regard, important atmospheric depositional events have been recorded in the study
alpine sites of southern Iberia from the mid-19th century to the first decades of the 20th
century caused by both enhanced North African dust fluxes (Mulitza et al., 2010)
(Jiménez et al., 2018) as well as a spike in atmospheric pollution (as observed in
anthropogenic Pb and Hg records in Sierra Nevada; Fig. 8) (Garcia-Alix et al., 2017;
Garcia-Alix et al., 2013). Similarly, both phenomena have been demonstrated as triggers
for glacier retreat (Painter et al., 2013) and snow melt in the Alps (Di Mauro et al., 2018).

Melting of the last glaciated area in the Sierra Nevada during the first decades of
the 20$^{st}$ century (Grunewald and Scheithauer, 2010) represents an important turning point
regarding recent environmental change in this alpine region (Garcia-Alix et al., 2017;
Jiménez et al., 2019). The rapid pace of environmental change in the area after this date
is attributed to an amplified effect of warming and aridification (Fig. 8b,c) that increased
stress on vulnerable ecosystems (Garcia-Alix et al., 2017; Jiménez et al., 2019; Jiménez
et al., 2018) with little hope for return of local glaciers.

**5. Concluding remarks**

This study shows the vulnerability of alpine regions and the importance of their
monitoring for a better understanding of climate variability and future rapid responses. In
this regard, algal-derived biomarkers from LdRS records have given rise to the first long-
chain alky diol temperature calibration in freshwater environments by means of the
comparison with instrumental temperature time-series. The combination of both short and
long sediment cores has provided both a highly accurate LCD-temperature calibration for
the instrumental period and a long-term historical perspective on the modern warming.
This approach delivers a better time-integrated temperature model than discrete
temperature measurements for the 20$^{th}$ century. Nevertheless, the lack of information
about the biological sources of LCDs in the Sierra Nevada means that this calibration can
only be potentially applied to other lakes with a similar LCD distribution or in the same
alpine area.

The low sample resolution in the longer core before ~1500 CE precludes us from
constraining the main natural controls on temperatures in this high-elevation site for the
Common Era. However, the general trends support that the presumed primary effect of
greenhouse gases on temperatures reconstructed from algal-lipids in this alpine region of
southern Iberia is likely modulated by long-term solar forcing. In recent times,
greenhouse gases seem to be the major temperature driver in this high elevation site.
Volcanic forcing appears to have little effect on reconstructed temperatures in this alpine
area. The Atlantic Multidecadal Oscillation (AMO) have also shown to have a long-term
effect in the study area; however, due its complex nature, the real effect of the AMO on
LCD-reconstructed temperatures in LdRS records cannot be fully constrained. In any
case, the effect of the internal climate variability on local temperatures cannot be ruled
out. LdRS records also highlight the potential impact that non-climatic environmental
drivers such as atmospheric dust and pollution deposition can have exerted on these
remote alpine environments (e.g., glacier melting).

Alpine temperatures of southern Iberia exceeded the highest scores reached during
pre-industrial times in the 1950s. This means that the rate of warming throughout the 20th
century doubled that of the last stages of the LIA. Furthermore, this modern warming rate
is higher in the Sierra Nevada than in the Alps, pointing towards more environmental
stress in the Sierra Nevada ecosystems. In addition to the amplified effect of warming
and aridification, the local environmental pressure may have enhanced throughout the
20th century due to the disappearance of perennial snow fields and the gradual reduction
of the seasonal snow cover affecting the local water availability. Future projections
suggest that warming in this fragile alpine region will continue at similar rates or even
higher than ones registered during the last century.

**Data availability.** Fractional abundances of the $C_{28}$ 1,13-diol, $C_{28}$ 1,14-diol, $C_{30}$ 1,13-
diol, $C_{30}$ 1,14-diol, $C_{30}$ 1,15-diol, and $C_{32}$ 1,15-diol from the studied cores (LdRS shc and
LdRS lgc), along with the four reference temperature time-series at 3020 masl from 1908
to 2008 are available online at: https://issues.pangaea.de/browse/PDI-22604

**Supplement.** The supplement related to this article is available online at:
*https://doi.org/*

**Author contributions.** The study was conceived by AG-A and JLT. CPM, LJ, GJM,
and RSA recovered the sediment cores. AG-A analysed the samples and processed the
data. All co-authors discussed the data and equally contributed to the preparation of the
manuscript.

**Competing interests.** The authors declare that they have no conflict of interest.

**Acknowledgements.** We would like to thank V. Slaymark (University of Glasgow), for
her help preparing and analysing the organic and inorganic samples, as well as F.J. Bonet
García, C. González Hidalgo, and M.J. Esteban Parra for providing the temperature time-
series from the Sierra Nevada and the South of Spain. We also want to thank the Editor,
E. McClymont, and two anonymous reviewers for their useful comments and suggestions
that improved the manuscript.

**Financial support.** This study was supported by the project P11-RNM 7332 of the "Junta
de Andalucía", the projects CGL2017-85415-R, CGL2013-47038-R and CGL2011-
23483 of the "Ministerio de Economía y Competitividad of Spain and Fondo Europeo de
Desarrollo Regional FEDER", the project 87/2007 of the OAPN-Ministerio de Medio
Ambiente, as well as the research group RNM-190 (Junta de Andalucía). A.G.-A. was
also supported by a Marie Curie Intra-European Fellowship of the 7th Framework
Programme for Research, Technological Development and Demonstration of the
European Commission (NAOSIPUK. Grant Number: PIEF-GA-2012-623027) and by a
Ramón y Cajal Fellowship RYC-2015-18966 of the Spanish Government (Ministerio de
Economía y Competividad). J.L.T. hosted the NAOSIPUK project (PIEF-GA-2012-
623027) at the University of Glasgow.

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

**FIGURES**

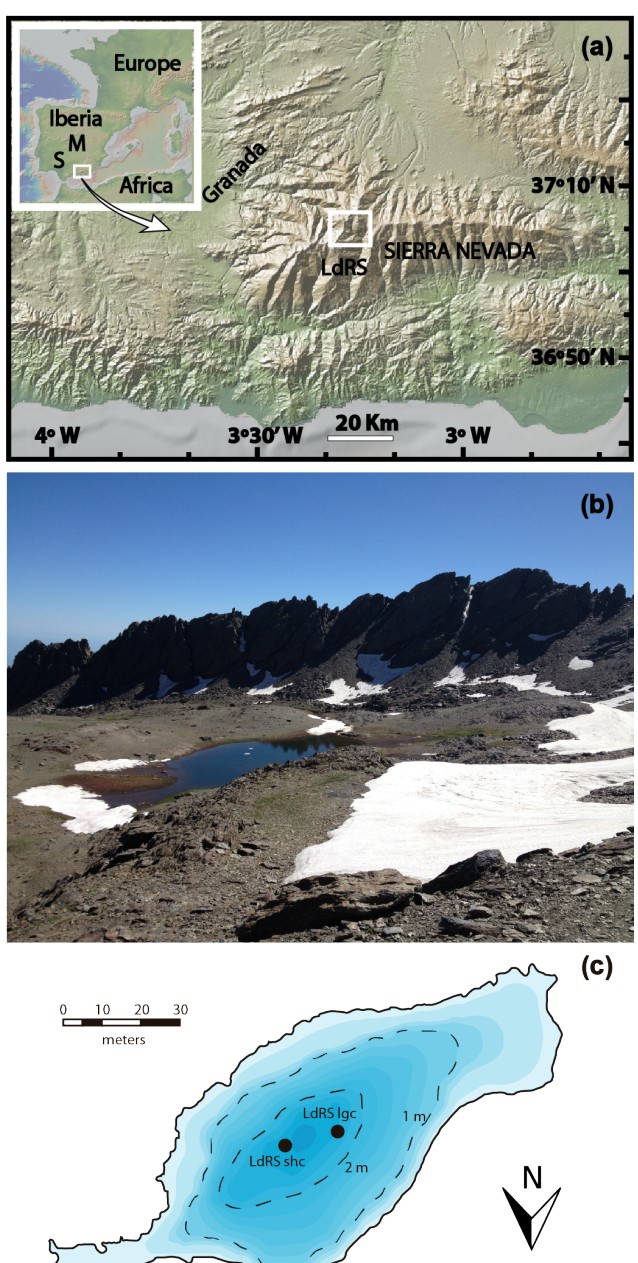


**Figure 1. Geographical setting.** (**a**) Location of the Sierra Nevada in the western
Mediterranean region, Madrid (M), Sevilla (S) and Granada observatories, as well as the
studied area: Laguna de Río Seco (LdRS), (**b**) LdRS catchment basin (0.42 ha) in spring
2013, (**c**) bathymetry map of LdRS along with the sampling points of both cores. Data
source and software: (**a**) map created by A. García-Alix using GeoMapApp (3.6.6)
(http://www.geomapapp.org), (**b**) picture from A. García-Alix, (**c**) digitalized map of a
bathymetry report from Egmasa S.A.

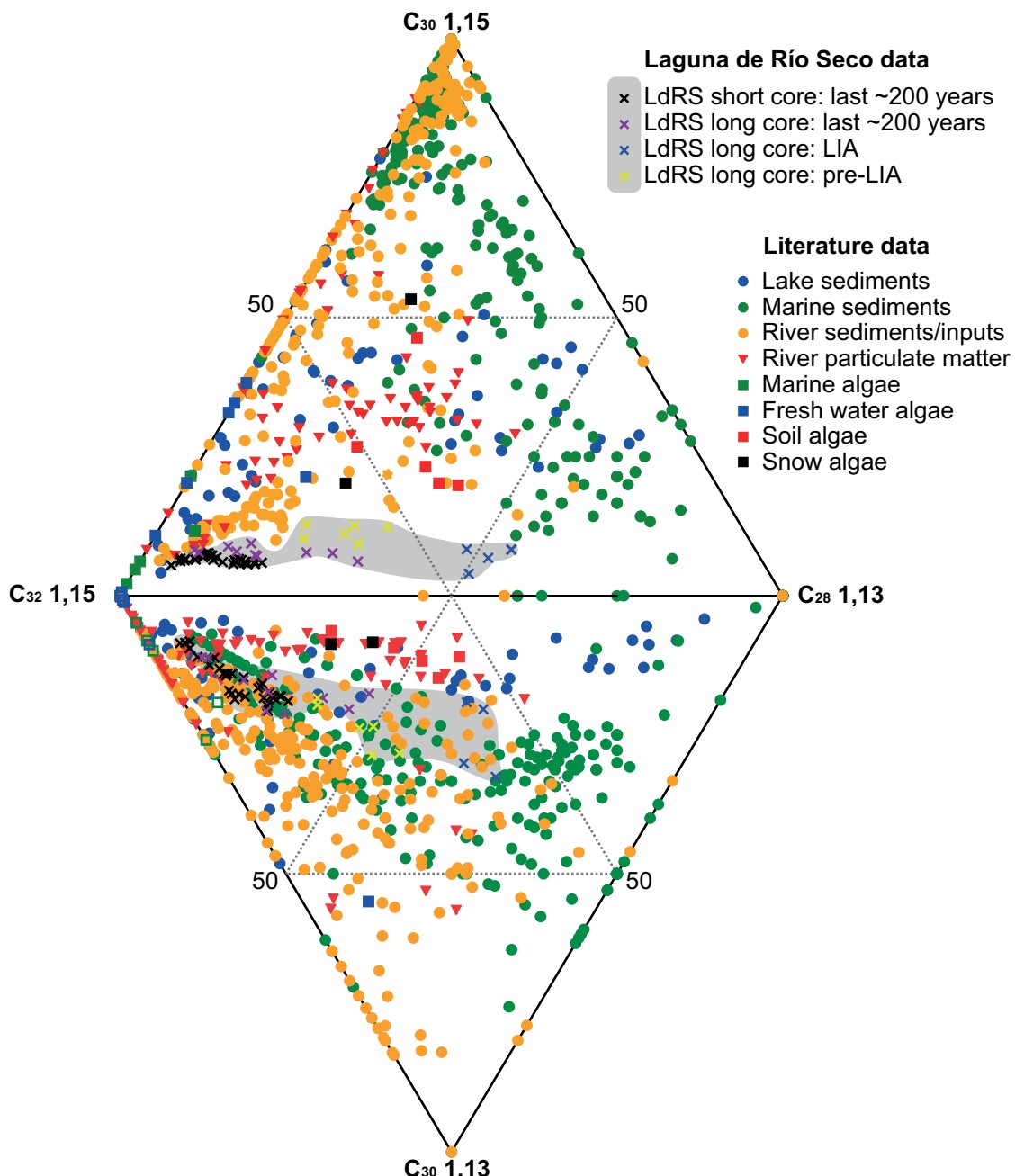


**Figure 2.** Double-ternary diagram of the relative abundances of $C_{28}$ 1,13-diol, $C_{30}$ 1,13-diol, $C_{30}$ 1,15-diol, and $C_{32}$ 1,15-diol from LdRS short core (LdRS shc ~200 years) and LdRS long core (LdRS lgc ~1500 years). Diol data compiled from the literature: lake sediments (Rampen et al., 2014a), algal cultures (Rampen et al., 2014a), marine sediments (de Bar et al., 2016; Lattaud et al., 2017a; Rampen et al., 2014b; Rampen et al., 2012), river sediments/inputs (de Bar et al., 2016; Lattaud et al., 2017b), river particulate organic matter (Lattaud et al., 2018a).

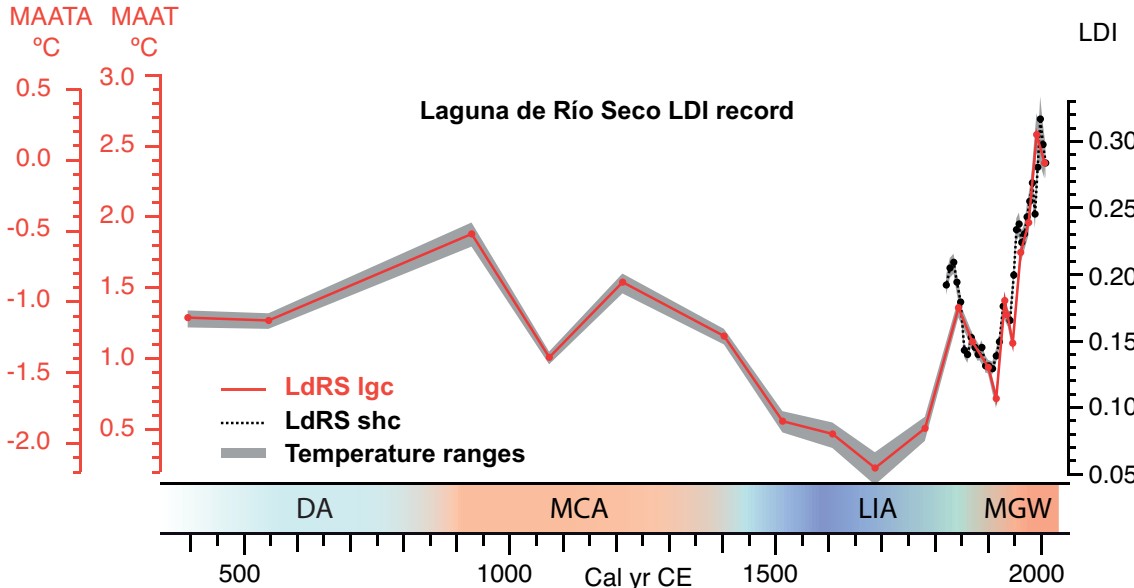


**Figure 3**. LDI record from LdRS, including both long core (solid line) and short core
(dashed line), mean annual air temperature (MAAT ºC) reconstruction from LDI records
of LdRS, as well as mean annual air temperature anomaly reconstruction (MAATAºC)
respect to the annual MAAT of the last 30 years (1979-2008). The grey shade shows the
reconstructed maximum and minimum temperature ranges obtained from the four LDI
individual calibrations.

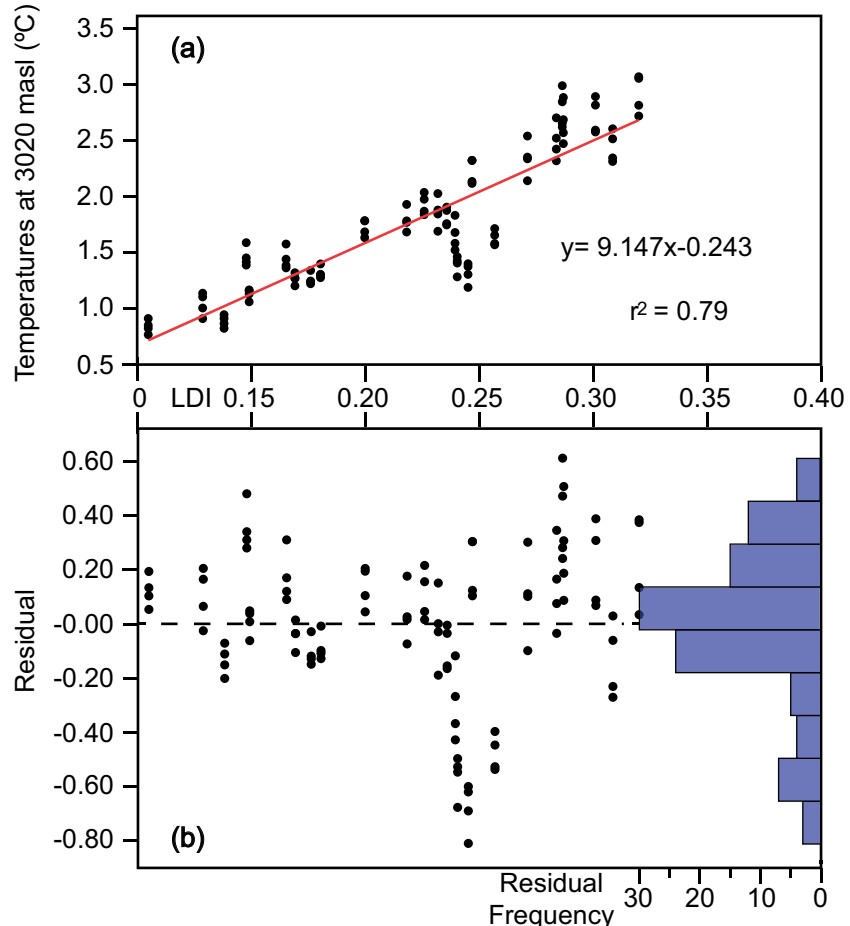

**Figure 4. LDI temperature calibration**. **(a)** Correlations by means of ordinary least
square regression between the both LDI records from LdRS (from 1908 to 2008) and the
four reference temperature time-series at 3020 masl. **(b)** LDI values of LdRS *vs* residual
temperatures (calculated between the calibrated LDI temperatures *vs* reference
temperature time series at 3020 masl), as well as the histogram of the frequency of these
residuals.

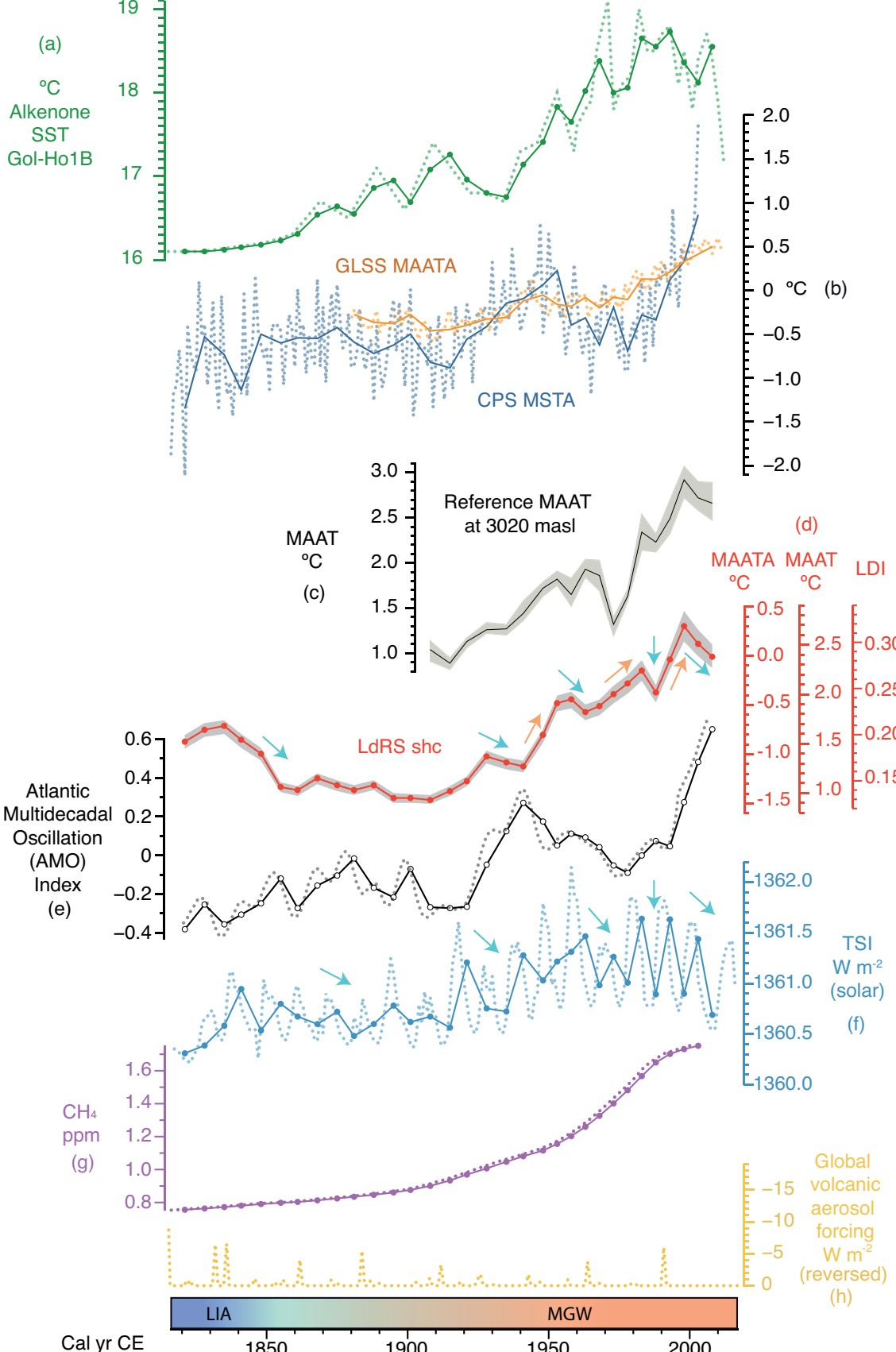


**Figure 5. Comparison of the LDI record and the reconstructed temperatures for the**
**last ~200 years of LdRS with marine and terrestrial temperature records, Atlantic**
**multidecadal oscillations, greenhouse gases, solar radiation and volcanic eruption**
**records.** Original data are in dashed lines. Solid lines represent the same time averaging
as the LDI data in LdRS shc (data were linearly interpolated and time-averaged to the
same resolution as the sampling points of LdRS shc) to facilitate the correlation. **(a)**
Alkenone-derived Sea Surface Temperatures (SST, ºC) of the core Gol-Ho1B_KSGC-31
(Gulf of Lion: NW Mediterranean Sea (Sicre et al., 2016)), **(b)** Composite-plus-scaling
(CPS) mean summer temperature anomaly reconstruction from tree rings records in
Europe with respect to 1974-2003 (MSTA ºC) (Luterbacher et al., 2016) as well as global
land and sea surface (GLSS) mean annual temperature anomalies with respect to 1979-
2008 CE (MAATA ºC) (Hansen et al., 2010), (c) Summary of the four reference
temperature time-series at 3020 masl: grey shade show the maximum and minimum
temperature range and the black solid line represent the mean temperature values, **(d)** LDI
record along with reconstructed mean annual air temperatures (MAAT ºC) and mean
annual air temperature anomalies with respect to 1979-2008 CE (MAATA ºC) for the last
~200 years in LdRS, **(e)** Atlantic Multidecadal Oscillation (AMO) reconstruction (Mann
et al., 2009), **(f)** high resolution total solar irradiance reconstruction (TSI, W m$^{-2}$)
(Coddington et al., 2016), **(g)** reconstructed concentration of atmospheric CH$_4$ (ppm)
(Schmidt et al., 2011), and **(h)** reconstruction of the global volcanic aerosol forcing (W
m$^{-2}$) (reversed) (Sigl et al., 2015). Acronyms: LIA, Little Ice Age; MGW, Modern Global
Warming. Blue arrows: decrease; orange arrows: increase.

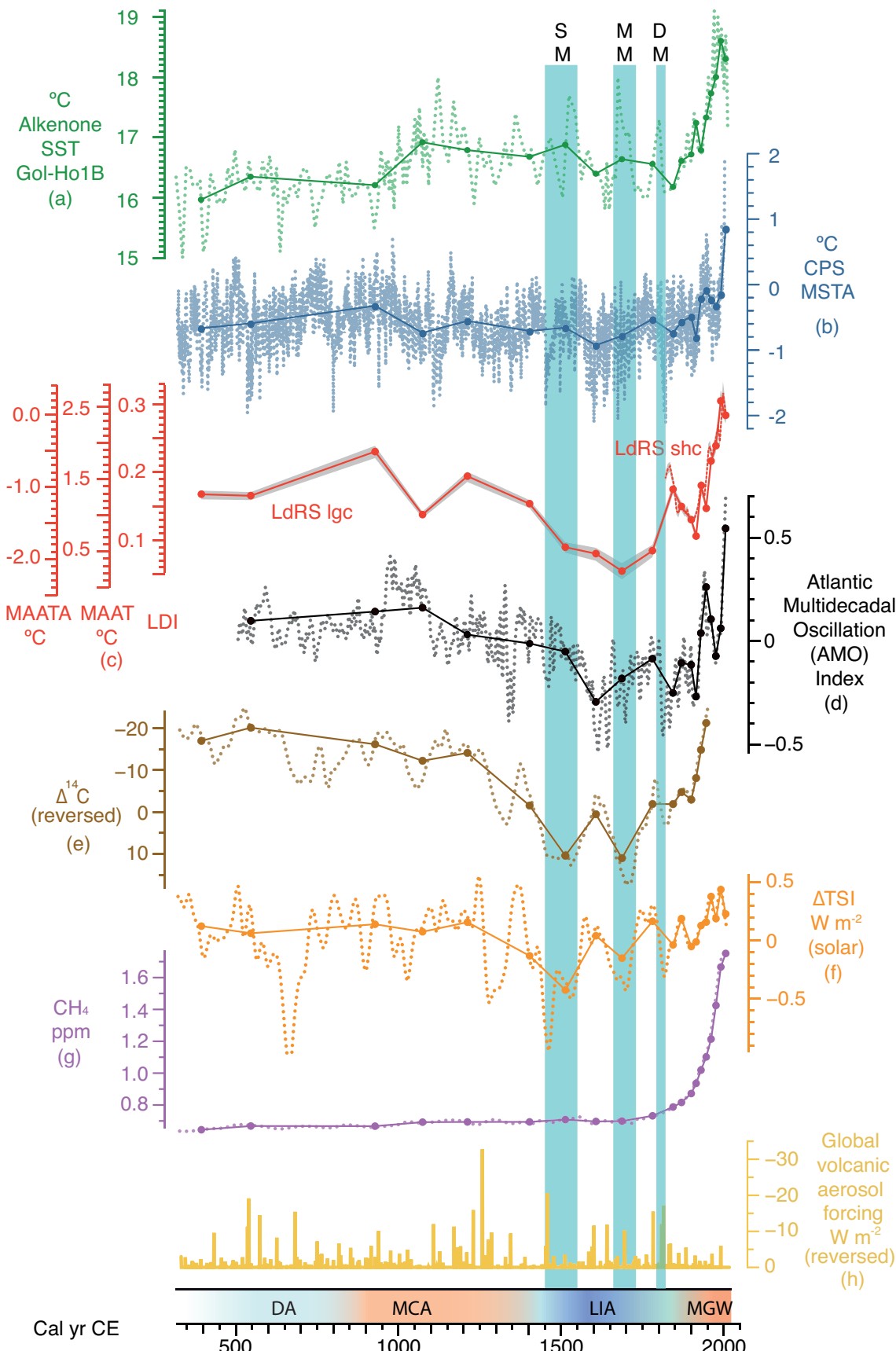

**Figure 6. Comparison of the LDI record and the reconstructed temperatures for the**
**last ~1500 years of LdRS with marine and terrestrial temperature records, Atlantic**
**multidecadal oscillations, solar radiation, greenhouse gases, and volcanic eruption**
**records.** Original data are in dashed lines. Solid dots represent the same time averaging
as the LDI data in LdRS lgc (data were linearly interpolated and time-averaged to the
same resolution as the sampling points of LdRS lgc) to facilitate the Pearson correlation:
**(a)** Alkenone-Sea Surface Temperatures (SST, ºC) of the core Gol-Ho1B_KSGC-31
(Gulf of Lion: NW Mediterranean Sea (Sicre et al., 2016)), **(b)** Composite-plus-scaling
(CPS) mean summer temperature anomaly reconstruction from tree rings records in
Europe with respect to 1974-2003 CE (MSTA ºC) (Luterbacher et al., 2016), **(c)** LDI
record along with reconstructed mean annual air temperatures (MAAT ºC) and mean
annual air temperature anomalies with respect to 1979-2008 CE (MAATA ºC) for the last
1500 years in LdRS, **(d)** Atlantic Multidecadal Oscillation (AMO) reconstruction (Mann
et al., 2009), **(e)** $\Delta^{14}C$ in the atmosphere (reversed) (Reimer et al., 2013), **(f)** reconstruction
of the difference of the total solar irradiance from the value of the PMOD composite
series during the solar cycle minimum of the year 1986 CE (1365.57 W m$^{-2}$) ($\Delta$TSI)
(Steinhilber et al., 2009), **(g)** reconstructed concentration of atmospheric $CH_4$ (ppm)
(Schmidt et al., 2011), and **(h)** reconstruction of the global volcanic aerosol forcing (W
m$^{-2}$) (reversed) (Sigl et al., 2015). Acronyms: DA, Dark Ages; MCA, Medieval Climate
Anomaly; LIA, Little Ice Age; MGW, Modern Global Warming. Blue bars show three
low solar activity periods, the Spörer Minimum (SM), the Maunder Minimum (MM), and
the Dalton Minimum (DM).


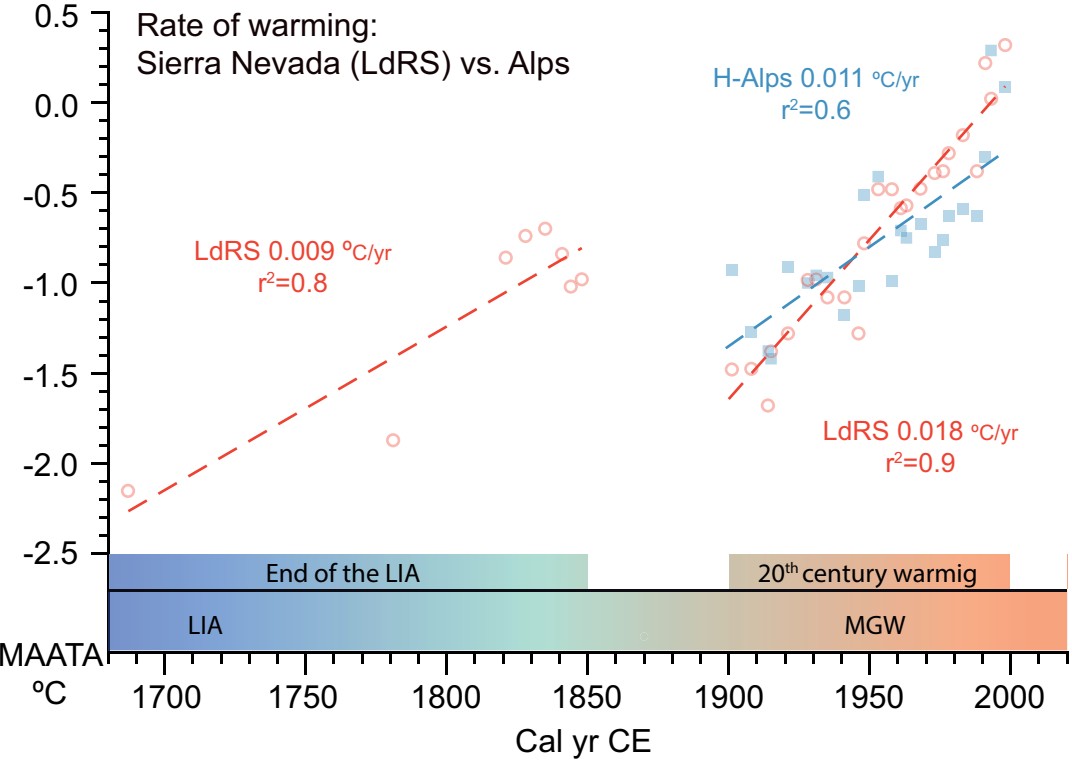


**Figure 7. Comparison between the average temperature warming rates from LdRS and the alpine areas of the Alps by means of ordinary least square regressions.** LDI-deduced MAATA (respect to the period 1979-2008 CE) from LdRS long and short cores for the last stage of the LIA and the 20th century (red open circles), and high-Alps historical (homogenised) temperature records from the Historical Instrumental Climatological Surface Time Series of the Greater Alpine Region (HISTALP) database (Auer et al., 2007; Böhm et al., 2010) at the same time averaging as LdRS shc-lgc to facilitate the comparison (blue closed squares).

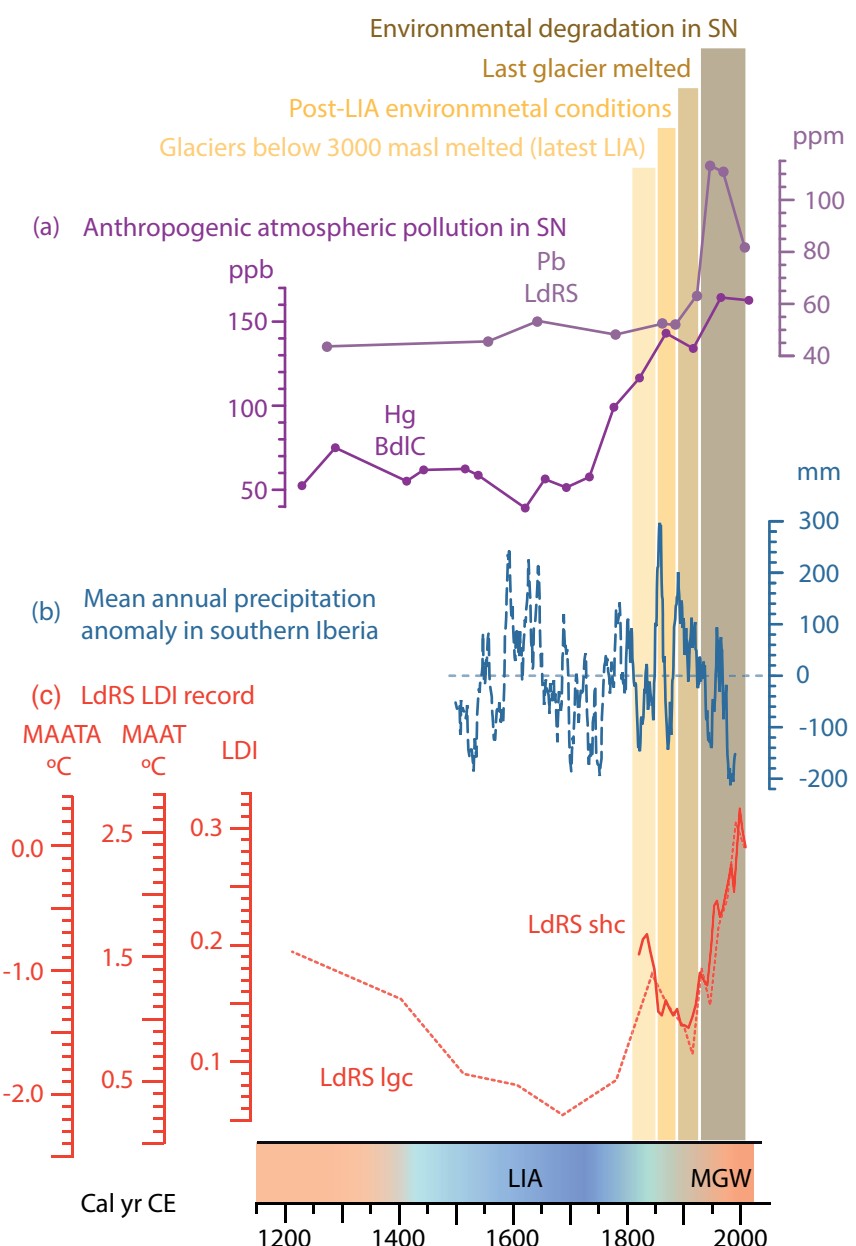

**Figure 8. Comparison among different factors affecting the environmental evolution of alpine wetlands in the Sierra Nevada. (a)** records of anthropogenic heavy metal atmospheric pollution (Pb and Hg) in two alpine sites of the Sierra Nevada: Laguna de Río Seco (LdRS) and Borreguil de la Caldera (BdlC) (Garcia-Alix et al., 2017; Garcia-Alix et al., 2013), **(b)** mean annual precipitation anomaly in southern Iberia from 1500 to 1990 CE with respect to the mean value of the instrumental period (1791-1990 CE): solid line- instrumental data from Gibraltar (southern Iberia); dashed line- anomaly precipitation reconstruction (Rodrigo et al., 1999), **(c)** LDI and reconstructed

temperatures in LdRS. Colour bars indicate the four main environmental stages in the

Sierra Nevada (SN) during the last 200 years. Acronyms: LIA, Little Ice Age; MGW,

Modern Global Warming.