# Peer review of "Algal lipids reveal unprecedented warming rates in alpine areas of SW Europe"

_Climate of the Past, 2019_

## Referee Comment (RC1) · Anonymous Referee #1 · 13 Sep 2019

In this paper, the authors use a novel proxy, based on long chain diols (LCDs), to reconstruct temperatures for the last 1500 years in an alpine lake from the southern Iberian Peninsula. Based on the reconstructed temperatures, the authors discuss the effect of greenhouse gasses and other climate affecting factors on the temperatures in alpine areas in SW Europe, and make predictions on what the temperatures and effects will be like at the end of the 21st century.

.

The authors present an interesting dataset from an exciting area. To the best of my knowledge, this is the first time LCDs have been used for temperature reconstruction in a freshwater environment, and the results are promising. However, since this is the first application of LCDs as a freshwater temperature proxy, I would expect a more

thorough discussion on this application, also because Rampen et al. (2014a) were critical in their study on the application of LCDs, and in particular on the Long chain Diol Index (LDI), as a freshwater temperature proxy.

.

In their marine LCD study, Rampen et al. (2012) observed a positive correlation between temperatures and the fractional abundances of C30 1,15-diol, a negative correlation between temperatures and the fractional abundances of C28 and C30 1,13-diol, and no correlation between temperatures and the fractional abundances of C32 1,15-diol. Moreover, they observed that the fractional abundance of C32 1,15-diol remained below 0.3 for most of the marine sediments. Based on those results, Rampen et al. (2012) introduced the LDI, with a stronger SST correlation compared to the fractional abundances of the individual LCDs.

Assuming the LDI is correlated with temperature in the studied lake Lago de Rio Seco (LdRS), the fractional abundances of the 1,13-diols do show negative correlations with temperature, but both the variation and the temperature correlation for C30 1,15-diol is extremely low, whereas the C32 1,15-diol does show a significant correlation over a wide range of fractional abundances. This means that the rationale behind the LDI in marine environments may not apply for LdRS. As a result, I would recommend to (also) test other LCD indices that include the C32 1,15-diol and/or multilinear regression analysis.

In addition, it seems like the correlations between the LDI and the fractional abundances of the individual LCDs are statistically different for the two sediment cores. With the exception for C30 1,13-diol, the slopes for the fractional-abundance-of-individual-diols from the 2 different cores differ significantly, when plotted vs the LDI.

The calibration of the LDI seems to be based on samples from 1908 and younger, and only using those samples obtained from the short core. Almost all of the fractional abundances of the 1,13-diols in the older sediments are higher, and almost all of the

fractional abundances of the C32 1,15-diol in the older sediments are lower. As a result, the reconstructed temperatures before 1908 are a result of extrapolation of the dataset, and one might even argue that the LCD distribution was significantly different in the samples before 1908.

In particular the C28 1,13-diol and C32 1,15-diol values show very different trends between the short and the long core for the overlapping time-period - the long core shows much larger ranges of values, something not mentioned in the manuscript. The different LCD distributions in the two cores for the overlapping period raises questions if a calibration, only based on samples from the short core, is applicable for the long core. What could possibly explain the (significant) differences between the two cores? Why were samples from the long core not included in the calibration, even though a number of samples fall in the time-period for which temperatures are available?

For these reasons, I consider it questionable if the authors provide sufficient support for the use of the LDI (or LCDs in general) for the temperature reconstruction in this study. To me, a better and more critical discussion seems crucial for this paper, also because the authors never seem to question their results and don't refrain from making some very strong statements, based on these results.

. .

Nit-picking and other comments. .

The title says "Extreme warming rates affecting alpine areas. . ." However, is it the extreme warming rates, or is it the extreme warming itself, that affect the alpine areas?

.

Lines 81-82. Rodrigo-Gámiz et al. (2015) tested the use of the LDI, but did not apply it as a temperature proxy - they did not perform climate reconstructions. Rampen et al. (2014b) also tested the applicability of long chain diols as temperature proxies without applying it for climate reconstruction. Rampen et al. (2014b) tested different indices

than the LDI - indices based on 1,14-diols.

.

Lines 82-83. Rampen et al. (2014a) tested the applicability of the LDI in freshwater environments, but did not use this proxy for climate reconstruction - to the best of my knowledge, no-one has published the use of LCDs for temperature reconstruction in freshwater environments so far.

.

Line 86. In my opinion, Rampen et al. (EPSL 276, p. 207-213, 2008) and/or Willmott et al. (Antarctic Science 22. P. 3-10, 2010) would be better references than De Bar et al. (2016), as they introduced and first applied the indices also used by De Bar et al. (2016) and others.

.

Lines 88-91. Are the authors specifically referring to paleoenvironmental reconstructions in freshwater environments here? Otherwise, I think the text and selected references do not do justice to the number of LCD studies that appeared recently.

Rampen et al. (2014a) did not apply LCDs for palaeoenvironmental reconstruction - they only tested the applicability of LCDs in freshwater environments.

.

Line 101. It is incorrect to state that the LDI has only been calibrated with other indirect temperature proxies - Rampen et al. (2014a) also correlated the LDI with annual mean air temperatures obtained from climate observation stations nearby the various lakes that were studied (e.g. see Fig. 5 in their paper)

.

Lines 102-104. Unless they provide reasons to believe otherwise, the authors should

emphasize that for now, their calibration is only applicable for LdRS.

.

Lines 170-173. The difference in the sedimentation rates above 16cm (0.13-0.9 cm/yr) and below (~0.008 cm/yr) seems large and relevant to me. I think the reason for this change in sediment rates, and the possible effects for this study, should be discussed.

.

Lines 264-267. Isn't this an indication that the LDI calibration from this study cannot be directly applied for other LCD studies?

.

Lines 274-276. It is unclear why a correlation between the LDI and the abundance of Chrysophyceae cysts would be an indication that these algae could be the source of the LCDs; to me, this only seems to indicate that Chrysophyceae are more abundant during warmer periods. The LDI is a ratio between various LCDs and should be independent from the abundance of their source organisms, unless LCDs have multiple sources, and specific LCDs are produced by specific organisms. It would be more relevant if a correlation between the absolute abundances of LCDs and algal numbers was observed.

.

Lines 297-299. Why is the calibration only based on LCD data from the sort core? How was dealt with the fact that samples may contain a signal collected over several years - how was the instrumental temperature selected?

I'm not convinced it is correct that only one regression analysis is performed in which, for each sediment sample, four different temperatures are included; every sample appears four times in this regression analysis. I think it would be better to perform four different regression analyses; one for each of the four reference temperature series.

It would be useful if the reference temperature data was also provided, for example in table S7. I would like to see the instrumental temperatures in a figure, for example in figure 5.

.

Line 308. There are too many decimals indicated in this equation. Also, there are 19 samples (n=19) used for the calibration, not 76.

.

Line 319. As indicated above, I would start the discussion with a critical look at the LCD data.

.

Line 346. What about the prominent warming observed in the LDI around 1830, which is not registered in other records?

.

Line 441. I really don't think the resolution of the LCD record for the LIA is high enough to identify 'events'.

.

Lines 477-479. I think that three sample points in the period between 1690 to 1850 are insufficient to determine the warming rate for that time-period. "... a low sample density for the LIA, which might slightly increase the uncertainty for this period..." in lines 482-483 seems like a strong understatement to me; I would refrain from making statements based on this warming rate.

.

Line 488. To me, it is not clear why slower warming rates in other European alpine areas are indicated as "An even more alarming result". How do the slower warming

rates in other areas affect the Sierra Nevada?

.

Lines 497-498. I don't see how the 'limited' LCD data in this study can be directly applied for an extrapolation to predict temperatures for 100 years in the future. I don't think this is the correct way to make such statements; I believe climate prediction is a very complex study area, and the simplification demonstrated here is almost offensive to that particular field of research.

In contrast to this simple extrapolation of a trend observed over the last century, one can also claim that in 100 years the temperatures will be more than 3 °C cooler than now, as demonstrated by the trend that started at the beginning of the 21st century (like the warming rate for the last stages of the LIA, the cooling rate for the 21st century of $\sim$0.32 °C/decade is based on 3 data points). Climate predictions should not be that simple.

The lack of restraint to make such extrapolations, the lack of research to test these predictions, the lack of additional information (other studies) about climate predictions, and the lack of restraint to predict the effects of the possible future warming in a very vague and yet very alarmist way, without providing any other type of support, is not correct.

In my opinion, the text between lines 497-519 should be removed. In a way, it also affects the credibility for the rest of this paper.

Fig. 7c does not show that temperatures may rise at least $\sim$1.4 °C (strange annotation, 'at least' combined with $\sim$) by the end of the 21st century.

.

Lines 568-571. As indicated before, this is not the first study in which LCD temperatures were calibrated with instrumental data. This has already been done by Rampen et al. (2014a).

.

Line 592. According to figure 5C, there is no such thing as an abrupt temperature increase in 1950s; if anything, the warming trend briefly flattened during the 1950s. The warming started around ∼1900 and continued until ∼2000?

.

Figure 2b and c. It is unclear to me why these two figures cannot be combined in one. The lines and data points are exactly the same, the only difference is the scale on the y-axis.

.

Figure 5. I don't understand how some of the linearly interpolated data points can deviate that much from the original dataset. This is most clearly visible in 5a.

---

## Referee Comment (RC2) · Anonymous Referee #2 · 16 Sep 2019

García-Alix et al. studied long-chain diols extracted from sediments from 2 cores from Laguna de Rio Seco. Long chain diols are novel biomarkers in lacustrine environments that have not been used for paleo-reconstruction, but, from what is known from the marine realm, they could be a good paleo-thermometer. The authors used the LDI, which is the ratio of different long-chain diols, calculated following Rampen et al. (2012) and calibrated against air temperature from different instrumental stations from lower elevation corrected for altitude effect. From this calibration, based on the last 100 years of instrumental record, they extrapolate LDI temperature on the last 1500 years. From the diol distribution in LdRS the authors deduct a different source organism than though until then, i.e. freshwater eustigmatophytes.

Global comments: The study site is extremely interesting, as Mediterranean alpine en-

vironments are prone to rapid changes, and uncovering the causes of environmental changes in these high elevation sites would be helpful for understanding future climatic changes. Furthermore, long-chain diols are rarely used in lacustrine environments and developing a temperature calibration would be particularly interesting as long-chain diols are commonly found in lake sediments globally distributed (Rampen et al., 2014). However, Rampen et al., 2014 did a thorough study (n=62 lake sediments) of possible correlation of the LDI and diol fractional abundances with annual mean air temperatures and/or GDGT-reconstructed lake temperatures and concluded that LDI does not seem applicable in lake environments. As such, why is only LDI tested for temperature and not any other diol fractional abundances mentioned in Rampen et al. (2014)? In particular, as the C32 1,15-diol seems to have a positive relation with temperature in cultures (as in Rampen et al., 2014, Goniochloropsis) and the author's dataset, why not test for the C32:0 1,15 over the C32:1 1,15? Only one m/z has to be added to the SIM mode (m/z=339). More details on seasonal temperature calibration would be interesting to mention as diol are subject to seasonality (Smith et al., 2013; Lattaud et al., 2018). Furthermore, the conclusion on the organisms producing the diols lacks concrete evidence, as the LDI (a ratio) gives no indication of diol-producer abundances. It would be better to compare the concentration of diols in the sediment with the number of cysts. As the lake studied is so specific a thorough study of all diol present are needed (especially if any source organisms is hypothesized) and should be reported at least as results such as 1,14-diols, C32 1,16-diol, C34 1,17 etc.

L27: what do you define as extreme responses?

L29: Rather than "algal lipids" the study calibrated algal lipid proxies

L30: Rephrase "extending alpine temperatures backward 1500 years", I suggest: "extending alpine temperature reconstructions to 1500 years before present"

L60: Instead of "this is the case", "as it is the case"

L87: Willmott et al., 2010 would be a reference to mention in term of nutrient proxies

L182-183: In Table S7 21 samples are reported for the long core.

L197: How was the concentration evaluated as no internal/external standard is mentioned?

L215: spacing between "from 1965 to 2011" and "(Spanish National..."

L218-220: Is there any GDGT/alkenones detected? As they could provide another independent temperature for calibration.

L265: "(C28, C30 and C32 1,13- and 1,15-diols)" do you find the C32 1,13-diol? Furthermore, there is no significant difference between the diol distribution of the short and long core recent samples (last 200y) and what has been previously published, that should be stated in the manuscript or a statistical test should be provided if the authors think otherwise. Only the samples from the LIA seems to fall close to the marine sediment distribution and might point toward a shift in producer but could also be an adaptation to the cold from the same organism, a more detailed discussion is needed.

L273: due to their small size (<3 um), eustigmatophytes are usually overlooked during planktonic study that does not include DNA sequencing. DNA sequencing on the modern lake water would bring stronger evidence.

L275: quite a bold statement without explanation in the text, explain the method to obtain the figure S3 (cyst count and identification). As the diol distribution is not significantly different from the previously published distribution a more thorough discussion is needed on why Chromulina spp. are potential diol producers and not freshwater eustigmatophytes. The comparison on fig S3 between LDI and Chromulina cyst is not an evidence as LDI do not correlate with diol abundance (it is actually independent), nor with diol-producer biomass (Balzano et al., 2016). Are any long-chain alcohols present? Or Long-chain ketones? As they would give idea on the producers (Volkman et al., 1999) and the possible state of degradation of the sediments (Versteegh et al., 2000).

L298: In figure 3 there is a group of points (LDI between 0.23-0.27) that deviates from the general correlation, are they all from the same period? Such as the LIA? If so, the LIA seem to be significantly different from the rest of the core and need to be handled independently.

L304-306: doing an outlier test would provide significance to this statement on the 1973 samples.

L350: Fig4 should be inversed with Fig5 as Fig5 is discussed before in the manuscript.

L352: The argument is reverse, the tree ring record supports the LDI data as it is a more known and used proxy.

L352-354: Are the warming rate from Southern Europe/Spain also stabilizing?

L433: The LDI record does not have a sufficient resolution to recognize a 1 year-long event.

L437-438: The cooling in the LDI of 1450-1500 and 1690 CE could also be attributed to solar minima rather than volcanic eruption. What about the volcanic aerosol from 1200-1300 CE that do not seem to impact the LDI in LdRS?

L473: Precise the number of samples analyzed for the MCA. The MCA baseline seems to be only represented by one samples, the rest of the MCA samples are much cooler. An average temperature of all the MCA samples is a better representation of the MCA temperature and can be used as MCA baseline.

L477: Precise the number of samples analyzed for the LIA

L497-498: Provide a reference for the statement: "Future scenarios are not optimistic for Sierra Nevada alpine areas either as projected temperature may rise at least ∼1.4 °C by the end of the 21st century"

L544: Is the temperature records mentioned from this study or from instrumental data, is there any precipitation reconstruction existing for this region?

Fig 2b: the dashed line is almost not visible, either change colors or thickness. Add the timing of LIA and MCA to the figure. Add the temperature records from the instrumental data to help comparison.

Fig S1: Correct "row" by "raw" in (a) (c) (e) (g) (i) and (k). Add unit for axis y

Fig S3: Please, add legend to the y axis. The authors use r and not r2 like in other figures, homogenize.

---

## Author Response (AR1)

Granada, 29th November 2019

Dear Dr. McClymont,

Thank you very much for giving us the opportunity to revise the manuscript. We think that this new version of the manuscript solves all (or at least most) of the concerns pointed out by both Reviewers and provides a more complete discussion about the potential application of long chain alkyl diols as freshwater paleothermometres. Editor and Reviewers would see that we have made a huge effort to improve the manuscript, and we hope that fulfils all your requirements.

Consequently, the length of some sections has slightly increased. These are the major changes in the manuscript: thorough explanation of the selection of reference temperature time-series and potential seasonal effects (section 2.3), discussion about the LCD distributions in LdRS, as well as their potential sources by means of a comparison with literature data (sections 3.1 and 4.1), extended LCD-temperature calibrations, comparisons, and discussion (four new individual temperature-LDI calibrations by means of ordinary least square regressions, another LDI-temperature calibration including the 4 temperature reference series, and three new potential calibrations based on multiple linear regression equations (MLR): section 3.2). All these changes in the main text come with three new supplementary figures, and seven new supplementary tables. We have also included additional data (Table S14) such as the fractional abundances of the $C_{28}$ and $C_{30}$ 1,14-diols from the short core and the reference temperature-time series at 3020 masl. However, other diol isomers were not included since they were either scarce/absent or not recorded in the Selected-Ion Monitoring mode with the retention time window to register the $C_{28}$-$C_{32}$ diols. In any case the mases of these other diol isomers (when recorded) were not very abundant in the TIC mode. Future analyses on suspended particulate matter and a complete year of sediment trapping would help better understand the whole LCD distribution.

We have clarified all these points in this new version. Additionally, we have made other minor changes in order to improve the manuscript. Changes in the manuscript are indicated in red font. In the attached file, in blue, we answer point by point the reviewers' questions.

Yours sincerely,

Antonio García-Alix
Departamento de Estratigrafía y Paleontología
University of Granada
Spain
(for the authors)

**Response to Anonymous Referee #1**

**Rev#1:** In this paper, the authors use a novel proxy, based on long chain diols (LCDs), to reconstruct temperatures for the last 1500 years in an alpine lake from the southern Iberian Peninsula. Based on the reconstructed temperatures, the authors discuss the effect of greenhouse gasses and other climate affecting factors on the temperatures in alpine areas in SW Europe, and make predictions on what the temperatures and effects will be like at the end of the 21st century. The authors present an interesting dataset from an exciting area. To the best of my knowledge, this is the first time LCDs have been used for temperature reconstruction in a freshwater environment, and the results are promising.

Thank you very much for your comments.

**Rev#1:** However, since this is the first application of LCDs as a freshwater temperature proxy, I would expect a more thorough discussion on this application, also because Rampen et al. (2014a) were critical in their study on the application of LCDs, and in particular on the Long chain Diol Index (LDI), as a freshwater temperature proxy. In their marine LCD study, Rampen et al. (2012) observed a positive correlation between temperatures and the fractional abundances of C30 1,15-diol, a negative correlation between temperatures and the fractional abundances of C28 and C30 1,13-diol, and no correlation between temperatures and the fractional abundances of C32 1,15- diol. Moreover, they observed that the fractional abundance of C32 1,15-diol remained below 0.3 for most of the marine sediments. Based on those results, Rampen et al. (2012) introduced the LDI, with a stronger SST correlation compared to the fractional abundances of the individual LCDs. Assuming the LDI is correlated with temperature in the studied lake Lago de Rio Seco (LdRS), the fractional abundances of the 1,13-diols do show negative correlations with temperature, but both the variation and the temperature correlation for C30 1,15-diol is extremely low, whereas the C32 1,15-diol does show a significant correlation over a wide range of fractional abundances. This means that the rationale behind the LDI in marine environments may not apply for LdRS. As a result, I would recommend to (also) test other LCD indices that include the C32 1,15-diol and/or multilinear regression analysis.

We agree, and in the new version of the manuscript we have included a discussion about the LDC distribution in both LdRS (sections 3.1 and 4.1) sediment cores and we have tried (and compared) different approaches for the LCD-temperature calibration (based on linear regressions, multiple regressions -including C32 1,15-diols-, etc.: sections 2.3 and 3.2 and new supplementary information (Figs. S3, S4, Tables S7, S8)). We have also included samples of the long core (from 1908 to 2008) in the calibration to make it stronger (see a full explanation in Rev#1 comment 10). Rev#1 will see in the new supplementary information of the paper that the different equations do show very good correlations with temperature (Fig S3 and Tables S7, S8), which support our interpretation about the relationship of LCDs and temperatures in this alpine lake. We

believe that we have discussed this matter deeply in this new version of the manuscript (section 3.2).

Regarding the potential use of the C30 1,15-diol in our calibration equations, we would like to mention that the discussion of the potential relationship between LCDs and lake temperatures in the paper from Rampen et al. (2014a) also pointed the scarce correlation between C30 1,15-diol and both mean annual temperatures and GDGT-derived temperatures. However, this isomer was eventually used in a multilinear regression equation along with C28 and C30 1,13-diols, and C30 1,15-diol, and even in the LDI calibration equation, showing good correlation in both cases ($r^2>0.64$ once one outlier was removed).

**Rev#1:** In addition, it seems like the correlations between the LDI and the fractional abundances of the individual LCDs are statistically different for the two sediment cores. With the exception for C30 1,13-diol, the slopes for the fractional-abundance-of-individual diols from the 2 different cores differ significantly, when plotted vs the LDI. The calibration of the LDI seems to be based on samples from 1908 and younger, and only using those samples obtained from the short core. Almost all of the fractional abundances of the 1,13-diols in the older sediments are higher, and almost all of the fractional abundances of the C32 1,15-diol in the older sediments are lower. As a result, the reconstructed temperatures before 1908 are a result of extrapolation of the dataset, and one might even argue that the LCD distribution was significantly different in the samples before 1908. In particular the C28 1,13-diol and C32 1,15-diol values show very different trends between the short and the long core for the overlapping time-period - the long core shows much larger ranges of values, something not mentioned in the manuscript. The different LCD distributions in the two cores for the overlapping period raises questions if a calibration, only based on samples from the short core, is applicable for the long core. What could possibly explain the (significant) differences between the two cores? Why were samples from the long core not included in the calibration, even though a number of samples fall in the time-period for which temperatures are available? For these reasons, I consider it questionable if the authors provide sufficient support for the use of the LDI (or LCDs in general) for the temperature reconstruction in this study. To me, a better and more critical discussion seems crucial for this paper, also because the authors never seem to question their results and don't refrain from making some very strong statements, based on these results.

The fractional abundances of the different isomers vary throughout time (the last 1500 years) because they are supposed to be influenced by changing environmental variables such as temperature. This may be one explanation for the comment "Almost all of the fractional abundances of the 1,13-diols in the older sediments are higher, and almost all of the fractional abundances of the C32 1,15-diol in the older sediments are lower": temperatures are different at the beginning (~1500 years ago) and at the end (2008 CE) of the core, and therefore, the fractional abundances of temperature-dependent isomers as

well. We have clarified the relationship between the different diol isomers and the temperatures for the last 1500 years in section 3.1.

Regarding the potential LCD differences between both cores, samples from both cores have different time averaging, and therefore fractional abundances for the same specific age (with different time averaging depending on the core) might be slightly different in each core. The location of the cores in different areas of the lake could also affect the registered LCD abundances. The use of ratios (i.e. LDI) in the calibrations may partially solve this issue. Anyway, it is worth mentioning that individual LCDs for the overlapping period in both cores show the same general trends. The comparison of the slopes is not the best way to compare both data sets, especially when the slopes are calculated from scarce samples, as it is the case of the overlapping period for the long core. Data from both cores can be compared using a Pearson correlation, but these data have to be converted to the same time-averaged age in order to perform a correlation between individual LCDs. In this case, the Pearson correlation (r) between both cores for the calibrated time period (1908-2008) is: 0.93 (C28 1,13-diol), 0.91 (C30 1,13-diol), 0.81 (C30 1,15-diol), and 0.91 (C32 1,15-diol), no p value is provided due to the low number of samples (n=7). If we consider the whole overlapping period, the Pearson correlation (r) between both cores is: 0.86 (C28 1,13-diol), 0.81 (C30 1,13-diol), 0.90 (C30 1,15-diol), and 0.89 (C32 1,15-diol) (n=10).

Another concern of Rev#1 was: "*The different LCD distributions in the two cores for the overlapping period raises questions if a calibration, only based on samples from the short core, is applicable for the long core*.". To fix this issue we have included the long core data in a new calibration, which is base in a total of 26 samples (short + long core samples). Both, the slope and the $r^2$ are pretty similar to those of the previous calibration (only using the 19 short core samples). Although the calibration is performed in down-core samples (from 2008 to 1908), the application of this new calibration is not an extrapolation of the dataset since the correlations are performed between diol indices and temperature data, without a time or depth constrain. Actually, temperatures just before 1900 slightly increased, even though the down-core trend from ~1950 to 1908 shows a decreasing temperature trend.

We thank Rev#1 for these comments that have helped us to improve the diol discussion in the manuscript.

**Rev#1:** Nit-picking and other comments.

1. **Rev#1:** The title says "Extreme warming rates affecting alpine areas. . ." However, is it the extreme warming rates, or is it the extreme warming itself, that affect the alpine areas?

Changes in the local flora and fauna in these ecosystems (Menéndez et al., 2014) are actually affected by both the extreme warming that is causing an enhanced melting of the permafrost and seasonal snow / ice, and the extreme rates of warming (i.e. almost 0.2

ºC/decade, which is higher than the global one according to the IPCC, 2013 of ~0.06 ºC/decade).

2. **Rev#1:** Lines 81-82. Rodrigo-Gámiz et al. (2015) tested the use of the LDI, but did not apply it as a temperature proxy - they did not perform climate reconstructions. Rampen et al. (2014b) also tested the applicability of long chain diols as temperature proxies without applying it for climate reconstruction. Rampen et al. (2014b) tested different indices than the LDI - indices based on 1,14-diols. AND Lines 82-83. Rampen et al. (2014a) tested the applicability of the LDI in freshwater environments, but did not use this proxy for climate reconstruction - to the best of my knowledge, no-one has published the use of LCDs for temperature reconstruction in freshwater environments so far.

We totally agree with Rev#1, and we have re-phrased the sentence: *"Another promising type of algal lipid biomarkers, the long-chain alkyl diols (hereafter LCDs), have also been assessed as temperature proxy in marine areas (Rampen et al., 2014b; Rampen et al., 2012; Rodrigo-Gámiz et al., 2014; Rodrigo-Gámiz et al., 2015). Nevertheless, the relationship between LCDs and temperature has only been tentatively tested in freshwater environments (Rampen et al., 2014a)."* Lines 80-84.

3. **Rev#1:** Line 86. In my opinion, Rampen et al. (EPSL 276, p. 207-213, 2008) and/or Willmott et al. (Antarctic Science 22. P. 3-10, 2010) would be better references than De Bar et al. (2016), as they introduced and first applied the indices also used by De Bar et al. (2016) and others.

Thanks for the suggestion - these references have been changed.

4. **Rev#1:** Lines 88-91. Are the authors specifically referring to paleoenvironmental reconstructions in freshwater environments here? Otherwise, I think the text and selected references do not do justice to the number of LCD studies that appeared recently. Rampen et al. (2014a) did not apply LCDs for palaeoenvironmental reconstruction - they only tested the applicability of LCDs in freshwater environments.

That is right. We referred to paleoenvironmental reconstruction in lakes. We have rephrased the sentence: "*Nevertheless, only a few studies have tested them as lacustrine archives of paleoproductivity (Shimokawara et al., 2010), past rainfall anomalies (Romero-Viana et al., 2012), or temperatures (Rampen et al., 2014a), among others.*" Lines 89-91.

5. **Rev#1:** Line 101. It is incorrect to state that the LDI has only been calibrated with other indirect temperature proxies - Rampen et al. (2014a) also correlated the LDI with annual mean air temperatures obtained from climate observation stations nearby the various lakes that were studied (e.g. see Fig. 5 in their paper).

Thank you for the remark. We have rephrased the sentence, including not only the LDI: "*The application of LCDs as a temperature proxy is novel in freshwater environments and only two preliminary calibrations based on recent surface sediments have been*

*obtained using both mean annual air temperatures (weather station data) and organic-derived temperature proxies (GDGTs) (Rampen et al., 2014a)"* Lines 101-104.

6. **Rev#1:** Lines 102-104. Unless they provide reasons to believe otherwise, the authors should emphasize that for now, their calibration is only applicable for LdRS.

We agree. However, this calibration although "local" could be applied to other alpine lakes in the Sierra Nevada, since alpine lakes in this area are pretty close and have similar algae communities (Barea-Arco et al., 2001; Sánchez-Castillo, 1988; Sánchez-Castillo et al., 1989). We have rephrased the paragraph: "*Although this calibration can only be applied to the studied lake at present, and perhaps to other alpine wetlands in the Sierra Nevada area, these new data support and reinforce the promising use of LCDs as a paleotemperature proxy in freshwater environments.*" Lines 107-110.

7. **Rev#1:** Lines 170-173. The difference in the sedimentation rates above 16cm (0.13-0.9 cm/yr) and below (0.008 cm/yr) seems large and relevant to me. I think the reason for this change in sediment rates, and the possible effects for this study, should be discussed.

Thank you for this comment, which made us notice that the 0.9cm/yr sedimentation rate was a typo. Actually, the sedimentary rate is between 0.09 and 0.13 cm/yr. We have corrected the typo. In any case, the main reason of a ~10 times increase in the sedimentation rate (from 0.008 to more than 0.09) is mainly due to snow/ice meltings during the last stages of the LIA and post-LIA, as well as human activities (i.e. construction of pathways, refuges) in the alpines areas of Sierra Nevada during the 19[th] century (García Montoro et al., 2016; Titos Martínez, 2019; Titos Martínez and Ramos Lafuente, 2016) that intensified after the 40s of the 20[th] century (Jiménez et al., 2015). Since the catchment basin of the lake is bare, with only few patches of vegetation surrounding the main water body (no nutrient supply), erosion mostly provided inorganic material (mica-schist and clays). The main effect of the high sedimentation rate on the algal community was the dilution of algal compounds such as chlorophylls and labile carotenoids, not affecting the relationship of these pigments with temperatures (Jiménez et al., 2015). We have explained so in the last part of the Introduction (new lines 141-150), and corrected the typo in the methodology section (line 187).

8. **Rev#1:** Lines 264-267. Isn't this an indication that the LDI calibration from this study cannot be directly applied for other LCD studies?

We agree and dealt with this comment in Rev#1 comment 6, but included a further new sentence in this paragraph remarking so: "*Consequently, the outcomes of this paper (i.e., the LCD-based temperature calibration) should not be generally applied to other freshwater records unless they show a similar LCD distribution as LdRS.*" Lines 522-525.

9. **Rev#1:** Lines 274-276. It is unclear why a correlation between the LDI and the abundance of Chrysophyceae cysts would be an indication that these algae could

be the source of the LCDs; to me, this only seems to indicate that Chrysophyceae are more abundant during warmer periods. The LDI is a ratio between various LCDs and should be independent from the abundance of their source organisms, unless LCDs have multiple sources, and specific LCDs are produced by specific organisms. It would be more relevant if a correlation between the absolute abundances of LCDs and algal numbers was observed.

We mentioned this hypothesis as an alternative to the potential biological sources since Eustigmatophyceae algae have not been identified in the alpine lakes of Sierra Nevada. However, both reviewers were concerned about the cyst and LDI relationship and therefore we opted to remove this sentence (former lines 274-276) and figure (Figure S3) from the manuscript. Further molecular and sediment traps studies (currently in progress) would be also required for this statement.

10. **Rev#1:** Lines 297-299. Why is the calibration only based on LCD data from the short core? How was dealt with the fact that samples may contain a signal collected over several years - how was the instrumental temperature selected?

We thought that samples from the same core would be better for a downcore calibration. However, Rev1# is right and including the samples from long core would mean more control points. Therefore, this time we have included 7 samples (from 2006 to 1908) from the long core in the calibration, and our new calibration is based on a total of 26 samples (see second comment for details).

Samples used in the calibration have a time averaging between 5 and 7 years; thus a mean of the historical temperatures covering the time averaging of each sample was done in order to have a mean temperature of the specific time-averaging of each sample. We had explained the issue in the captions of Figs., 4 and 5 i.e.: "*Solid dots represent the same time averaging as the LDI data in LdRS lgc*". However, we have also clarified this in section 3.2 lines 404-409.

The selection of the reference temperature time-series have been explained (and improved) in detail in the material and methods section *2.3 Reference temperature time-series for LDI temperature calibration*, taking into account the longest and more reliable temperature time-series in the area. Lines 235-351.

11. **Rev#1:** I'm not convinced it is correct that only one regression analysis is performed in which, for each sediment sample, four different temperatures are included; every sample appears four times in this regression analysis. I think it would be better to perform four different regression analyses; one for each of the four reference temperature series.

Yes, we agree and we have included all the possible combinations for the LDI temperature calibration: we have performed five LDI calibrations: 1) Ordinary Least Square Regressions between the two groups of reference temperature time-series at 3020 masl and the LDI record from LdRS shc and lgc, resulting in four individual calibration equations (Fig. S3), and 2) a calibration including the 4 temperature reference series vs.

the LDI (Fig. 4). The slopes of these four equations range from 8.2 to 10.2. The LDI-derived temperatures from the reference time-series 2 show the highest values for the last ~100 years, whereas the minimum values are mainly shown by the ones calculated with the reference time-series 1. The difference between the four LDI-derived temperatures for the last ~100 years is low, with a standard deviation lower than 0.13. The standard error of these four individual calibrations ranges from 0.18 to 0.23°C, and the maximum residual is ~0.8°C. However, due to the uncertainty of establishing an accurate temperature time-series at 3020 masl, LDI-derived temperature values from these LDI individual calibrations have been used to determine the range of the variation (minimum and maximum temperature values) for each point, and an additional calibration, summarising the relationship between LDI and the four reference temperatures at 3020 masl has been performed. The obtained 104 combinations of LDI and temperature data provided an equation representing the average relationship between MAAT and LDI (Eq. (2); Fig. 4a). So, in the figures, in addition to the LDI derived-temperatures from the LDI "average calibration", we are including the maximum and minimum ranges obtained from the four LDI individual calibrations (e.g., Fig. 3; Fig S4). In this way, in the same figure we are showing all the LDI-reconstructed temperatures (from the different calibrations) we obtained, allowing an easy (and more visual) comparison between the different environmental variables. We think that this is the best way to manage all these data. In addition to these LDI-temperature calibrations we have also discussed three models based on multiple linear regressions of the relative abundances of the differnet diol isomers, as Rev#2 requested. See all these changes in section 3.2-lines 404-482.

12. **Rev#1:** It would be useful if the reference temperature data was also provided, for example in table S7. I would like to see the instrumental temperatures in a figure, for example in figure 5.

Yes, we have included the mean instrumental temperature obtained from our four simulations in Fig. 5 (former Fig. 6) and the four reference temperature time-series in Supplementary Table S14.

13. **Rev#1:** Line 308. There are too many decimals indicated in this equation. Also, there are 19 samples (n=19) used for the calibration, not 76.

We have reduced the decimals and clarified the "n issue" in the text: There are now 26 samples (including those from the long core) and four temperature time-series: *"(n=26x4; $R^2$= 0.79)\**
*\* n = 26 LDI values plotted against four reference temperature time-series providing a total of 104 combinations"* lines 448-450.

14. **Rev#1:** Line 319. As indicated above, I would start the discussion with a critical look at the LCD data.

We have included a general discussion about the distribution of the different LCDs obtained and their potential temperature relationships (discussion section 4.1, lines 486-527).

15. **Rev#1:** Line 346. What about the prominent warming observed in the LDI around 1830, which is not registered in other records?

This warming follows the trend observed during the 18th century, and is related to the latest stages of the LIA. The study record comes from an alpine area, which is highly sensitive and may record this warming stronger than in other areas. In any case, the comparison with other records depends on the scale and time averaging. For example, the time averaging is high when comparing other records with the long core, which could prevent us from identifying specific events. However, looking carefully at this period (from ~1820 to 1840) in the high-resolution record of the short core (current Fig. 5), one can observe that the tree ring CPS MSTA record also registered this warming in the 1830s decade. In this paragraph we were describing the major trends for the last 1500 years and we did not pay attention in describing that in detail. However, this warming in the latest stages of the LIA was mentioned in section 4.4 (former section 4.3). In any case, we have tried to clarify this, and we have added a sentence in the paragraph that Rev#1 mentioned: *"Nevertheless, the warming documented from the LCD-derived temperatures in the last stages of the LIA is more pronounced in the LdRS record"* lines 556-557.

16. **Rev#1:** Line 441. I really don't think the resolution of the LCD record for the LIA is high enough to identify 'events'.

Rew#1 is partially right, we do not have enough sample resolution in the long core to perfectly track these events; however, the LdRS long core data during the LIA do not only represent specific temporal data, but time averaged samples. There is a mean time-averaging of ~87 years in each sample and thus we are not recording a specific moment, but a time-averaged period, where we see these temperature drops agreeing with periods of solar minima. Anyway, perhaps the word "correspond" in the text has too many connotations and we have rephrased the sentence, specifying that they are coeval only: *"Most of the above-mentioned cooling events recorded in LdRS, such as those during the LIA, are coeval with low solar activity periods…"* lines 656-657.

17. **Rev#1:** Lines 477-479. I think that three sample points in the period between 1690 to 1850 are insufficient to determine the warming rate for that time-period. ". . . a low sample density for the LIA, which might slightly increase the uncertainty for this period. . ." in lines 482-483 seems like a strong understatement to me; I would refrain from making statements based on this warming rate.

That is also partially right. Samples in the long core for the LIA have a time-averaging of ~87 years. Therefore, samples do not provide information about a specific moment, but a time-averaged period, what is more representative than a snapshot of a specific moment. In this new version we have also included samples from the short core for the first half of the 19th century (5 more samples: Fig. 7). We have explained these new data in section 4.4 lines 688-698.

18. **Rev#1:** Line 488. To me, it is not clear why slower warming rates in other European alpine areas are indicated as "An even more alarming result". How do the slower warming rates in other areas affect the Sierra Nevada?

We wanted to explain that it is alarming that Sierra Nevada had higher warming rates than the Alps during that period. We agree that the phrasing did not transmit this message. We thus have removed "An even more alarming result" in the revised version.

19. **Rev#1:** Lines 497-498. I don't see how the 'limited' LCD data in this study can be directly applied for an extrapolation to predict temperatures for 100 years in the future. I don't think this is the correct way to make such statements; I believe climate prediction is a very complex study area, and the simplification demonstrated here is almost offensive to that particular field of research. In contrast to this simple extrapolation of a trend observed over the last century, one can also claim that in 100 years the temperatures will be more than 3 ºC cooler than now, as demonstrated by the trend that started at the beginning of the 21st century (like the warming rate for the last stages of the LIA, the cooling rate for the 21st century of 0.32 ºC/decade is based on 3 data points). Climate predictions should not be that simple. The lack of restraint to make such extrapolations, the lack of research to test these predictions, the lack of additional information (other studies) about climate predictions, and the lack of restraint to predict the effects of the possible future warming in a very vague and yet very alarmist way, without providing any other type of support, is not correct. In my opinion, the text between lines 497-519 should be removed. In a way, it also affects the credibility for the rest of this paper.

We have removed this issue from the discussion, and added some sentences and references about published temperature projections in Sierra Nevada at ~1000 masl (at lower elevations than our study site). We have also explained the lack of these kind of projections in the alpine areas of the region, and the potential use of the our new data in the assessment of future scenarios. Lines 711-719.

20. **Rev#1:** Fig. 7c does not show that temperatures may rise at least 1.4 ºC (strange annotation, 'at least' combined with _) by the end of the 21st century.

We have removed this paragraph. See also comment Rev#1 19.

21. **Rev#1:** Lines 568-571. As indicated before, this is not the first study in which LCD temperatures were calibrated with instrumental data. This has already been done by Rampen et al. (2014a).

Rev#1 is right. We have changed the sentence clarifying that this is the first study using instrumental temperature time-series. Line 772.

22. **Rev#1:** Line 592. According to figure 5c, there is no such thing as an abrupt temperature increase in 1950s; if anything, the warming trend briefly flattened during the 1950s. The warming started around _1900 and continued until _2000?

Rev#1 is right. We only meant that alpine temperatures of southern Iberia exceeded the highest temperature scores reached during pre-industrial times in the 1950s. We have changed the sentence accordingly. Lines 796-797.

23. **Rev#1:** Figure 2b and c. It is unclear to me why these two figures cannot be combined in one. The lines and data points are exactly the same, the only difference is the scale on the y-axis.

We agree and we have combined both figures: Fig. 3.

24. **Rev#1:** Figure 5. I don't understand how some of the linearly interpolated data points can deviate that much from the original dataset. This is most clearly visible in 5a.

Dotted lines are only the lines connecting the original data points in Figs 4 and 5. The different environmental records have been lineally interpolated and time-averaged to the same intervals as the studied cores (dots/points) in order to properly compare them with LdRS data. For example, data of different records within the time interval 1942-1948 (as defined in the short core), have been time averaged (to the period 1942-1948) and assigned to the same age (i.e. 1948) to facilitate the correlation among them. Although this might create a slight offset in age with the original data, it is the best way to correlate all the records with the same time-averaged-intervals as the studied cores of LdRS.

**Response to Anonymous Referee #2**

**Rev#2:** García-Alix et al. studied long-chain diols extracted from sediments from 2 cores from Laguna de Rio Seco. Long chain diols are novel biomarkers in lacustrine environments that have not been used for paleo-reconstruction, but, from what is known from the marine realm, they could be a good paleo-thermometer. The authors used the LDI, which is the ratio of different long-chain diols, calculated following Rampen et al. (2012) and calibrated against air temperature from different instrumental stations from lower elevation corrected for altitude effect. From this calibration, based on the last 100 years of instrumental record, they extrapolate LDI temperature on the last 1500 years. From the diol distribution in LdRS the authors deduct a different source organism than though until then, i.e. freshwater eustigmatophytes.

**Rev#2:** Global comments: The study site is extremely interesting, as Mediterranean alpine environments are prone to rapid changes, and uncovering the causes of environmental changes in these high elevation sites would be helpful for understanding future climatic changes. Furthermore, long-chain diols are rarely used in lacustrine environments and developing a temperature calibration would be particularly interesting as long-chain diols are commonly found in lake sediments globally distributed (Rampen et al., 2014).

Thank you very much for your comments.

**Rev#2:** However, Rampen et al., 2014 did a thorough study (n=62 lake sediments) of possible correlation of the LDI and diol fractional abundances with annual mean air temperatures and/or GDGT-reconstructed lake temperatures and concluded that LDI does not seem applicable in lake environments. As such, why is only LDI tested for

temperature and not any other diol fractional abundances mentioned in Rampen et al. (2014)? In particular, as the C32 1,15-diol seems to have a positive relation with temperature in cultures (as in Rampen et al., 2014, Goniochloropsis) and the author's dataset, why not test for the C32:0 1,15 over the C32:1 1,15? Only one m/z has to be added to the SIM mode (m/z=339). More details on seasonal temperature calibration would be interesting to mention as diol are subject to seasonality (Smith et al., 2013; Lattaud et al., 2018). Furthermore, the conclusion on the organisms producing the diols lacks concrete evidence, as the LDI (a ratio) gives no indication of diol-producer abundances. It would be better to compare the concentration of diols in the sediment with the number of cysts. As the lake studied is so specific a thorough study of all diol present are needed (especially if any source organism is hypothesized) and should be reported at least as results such as 1,14-diols, C32 1,16-diol, C34 1,17 etc.

One of the main conclusions from Rampen et al. (2014a) was that although the relative abundances of individual LCDs in lakes did not show correlation with mean annual air temperatures, the GDGT deduced temperatures did show good correlation with the LDI ($r^2$=0.64). This correlation was even higher when using a multiple regression with the same LDI isomers ($r^2$=0.74). In both cases, one outlier of the dataset was removed to improve the correlation. Rampen et al. (2014a) also suggested that more tests are needed in freshwater environment to assess the application of LCDs in these environments and this is one of the main aims of our manuscript. In our paper we developed a LDI-temperature calibration because the relationship between LDI, and therefore the LCDs involved in the index, and instrumental temperatures was higher ($r^2$>0.8) than those of the different approaches performed in the above-mentioned paper. Anyway, Rev#2 is right and we are including a discussion of the distributions of the different LCDs identified (section 3.1), the different temperature calibrations based on different LCD indices: LDI, linear regressions, multiple regressions, etc (section 3.2, Figs 4, S3, S4, Table S8). We have also included some new tables in supplementary information showing these LCD-temperature correlations (Tables S3, S7)

We cannot re-run all the samples again, but we are now including different diols with the m/z that were previously measured, i.e. 1,14-diols. In addition, we did look for the C32:1 1,15-diol and the C32 1,16-diols in the TIC chromatograms, but they were not always present and in most of the samples the concentration was below the detection limit. This precluded us from getting any reliable data.

In the case of the short core we have provided the (very scarce) C28 and C30 1,14-diols, since these masses where previously measured in SIM mode (Table S14). In the case of the C34 1,17-diol, they were evaluated in the TIC mode (very low abundance), but we cannot provide their real relative abundances since the analyses with the Selected-Ion Monitoring mode (SIM) was only performed at the specific retention time window to identify the C28, C30, and C32 diols.

Regarding the relationship between LCDs and cyst, Rev#2 is right, this is not the best way to discuss about the producers. Further molecular and sediment traps studies (currently in progress) are also required to identify the biological source. We have changed this section (see also comment Rev#1 9).

As far as seasonality is concerned, we had slightly explained it in Table S3, S4 of the supplementary information. We agree with Rev#2, and we have discussed the potential seasonal issue in the new version of the main text in order to choose the bet reference temperature time-series for the LCD calibration (lines 267-303).

**Rev#2:** Nit-picking and other comments.

1. **Rev#2:** L27: what do you define as extreme responses?

We meant abrupt environmental responses, for example amplification of natural trends due to human pressure, i.e. Garcia-Alix et al. (2017). We did not explain that at this point since this is the abstract section. Anyway, we have changed the sentence: *"While major environmental shifts have occurred over the last ~1500 years in these alpine ecosystems, only changes in the recent centuries have led to abrupt environmental responses…"*

2. **Rev#2:** L29: Rather than "algal lipids" the study calibrated algal lipid proxies

Done. We have completed the sentence: "this study, for the first time, has calibrated an algal lipid-derived temperature proxy"

3. **Rev#2:** L30: Rephrase "extending alpine temperatures backward 1500 years", I suggest: "extending alpine temperature reconstructions to 1500 years before present"

Done.

4. **Rev#2:** L60: Instead of "this is the case", "as it is the case"

We have rephrased this sentence.

5. **Rev#2:** L87: Willmott et al., 2010 would be a reference to mention in term of nutrient proxies

Done. We have also included (Rampen et al., 2008) as Rev#1 suggested.

6. **Rev#2:** L182-183: In Table S7 21 samples are reported for the long core.

Rev#2 is right. We meant the samples that we really used. We have now mentioned the 21 samples in that sentence, and we have included a new sentence explaining that one of the samples fell below quantification limits: *"The sample at 19.5 cm depth in the long core was discarded because its concentration fell below quantification limits. "* lines 215-216.

7. **Rev#2:** L197: How was the concentration evaluated as no internal/external standard is mentioned?

In this step we only wanted to roughly know the concentration of the different compounds of the samples with the GC-FID so that the signal would not saturate the MS detector when we measured them in the GC-MS. We measured an external standard of cholesterol every five samples. We have included this explanation in the revised version. Lines 214-215.

8. **Rev#2:** L215: spacing between "from 1965 to 2011" and "(Spanish National: : :"
Done.

9. **Rev#2:** L218-220: Is there any GDGT/alkenones detected? As they could provide another independent temperature for calibration.
Sadly, our tests showed that there are not alkenones in the alpine lakes of the Sierra Nevada. The analyses of the polar fraction (to assess the GDGTs presence) is a new project that will develop once our equipment is properly set up for these kinds of heavy compounds.

10. **Rev#2:** L265: "(C28, C30 and C32 1,13- and 1,15-diols)" do you find the C32 1,13-diol? Furthermore, there is no significant difference between the diol distribution of the short and long core recent samples (last 200y) and what has been previously published, that should be stated in the manuscript or a statistical test should be provided if the authors think otherwise. Only the samples from the LIA seems to fall close to the marine sediment distribution and might point toward a shift in producer but could also be an adaptation to the cold from the same organism, a more detailed discussion is needed.
We did not find the C32 1,13-diol in our samples. In the new version we have specified "*C28 and C30 1,13-diols and C30 and C32 1,15-diols*". Following Rampen et al. (2014a), we had plotted *C28 1,13-diol, C30 1,13-diol, C30 1,15-diol and C32 1,15*-diol distributions in a ternary plot including the published diol distributions. We found that the LCD distribution in LdRS, even though close to that of riverine material, fitted with an almost blank region in the diagram published by Rampen et al. (2014a). In any case, we are including a brief discussion (section 4.1) about the LCD distributions of the different sources, along with a new double-ternary diagram (Fig. 2), Kruskal-Wallis ANOVA tests, and a Mann-Whitney U test (Table S9) to discuss about differences between the different diol sources (marine, lacustrine, algal cultures, etc). Lines 488-510.

11. **Rev#2:** L273: due to their small size (<3 um), eustigmatophytes are usually overlooked during planktonic study that does not include DNA sequencing. DNA sequencing on the modern lake water would bring stronger evidence.
Rev#2 is right. We are currently recovering material from sediment traps and suspended particulate material in order to conduct both geochemical and DNA sequencing analyses. This is an ongoing work, since this is an oligotrophic lake we would like to have at least two-annual cycles in order to catch up the different algal blooms and the LCDs producers and better understand the relationship between LCDs distribution and lake temperatures.

12. **Rev#2:** L275: quite a bold statement without explanation in the text, explain the method to obtain the figure S3 (cyst count and identification). As the diol distribution is not significantly different from the previously published distribution a more thorough discussion is needed on why Chromulina spp. are potential diol producers and not freshwater eustigmatophytes. The comparison on fig S3 between LDI and Chromulina cyst is not an evidence as LDI do not correlate with diol abundance (it is actually independent), nor with diol-producer biomass (Balzano et al., 2016). Are any long-chain alcohols present? Or Long-chain ketones? As they would give idea on the producers (Volkman et al., 1999) and the possible state of degradation of the sediments (Versteegh et al., 2000).

Rev#2 is right. See also response to comment 9 by Rev#1. We have removed this sentence regarding other potential biological producers since we do not have strong evidences of this fact. Regarding the other organic compounds, the algal productivity is not too high in this alpine oligotrophic lake. We have found *n*-alkanes and fatty acids in the sediments (from semi aquatic plants and from small plant patches in the catchment), but we have not found the shorter chains of these *n*-alkanes and fatty acids that might be related to algae/bacteria. We have not found long chain ketones either (see Rev#2 comment 9). Regarding the keto-ols occurrence, as possible degradation product of diols, we have only identified the C32 1,15-keto-ol (in TIC) in some of the samples, although in very low abundance. A further identification of these compounds would be very interesting for future analyses of current surface sediments, sediment traps, and suspended particulate matter (on going project: see Rev#2 comment 11).

13. **Rev#2:** L298: In figure 3 there is a group of points (LDI between 0.23-0.27) that deviates from the general correlation, are they all from the same period? Such as the LIA? If so, the LIA seem to be significantly different from the rest of the core and need to be handled independently.

In this figure we only showed data from 1908 to 2008 CE in order to calibrate the LDI vs historical temperatures. In the text we mentioned that the group of data referred by Rev#2 (corresponding to 1973: 1 LDI point vs. four temperature reconstructions at 3020 masl) might be an outlier.

14. **Rev#2:** L304-306: doing an outlier test would provide significance to this statement on the 1973 samples.

We have used the residuals in order to assess the outliers, since an outlier is a point with an unusually large residual, at least 2.5 standard deviations from the mean value (mean annual temperatures in our case). Data for 1973 show residuals that are 2.5 times higher than the standard deviations for one temperature reconstructions. This is why we suggested a potential outlier. We have clarified this in the result section. Line 444-446.

15. **Rev#2:** L350: Fig4 should be inversed with Fig5 as Fig5 is discussed before in the manuscript.

Figures 4 and 5 were mentioned at the same time in the manuscript, but Rev#2 may be right and the short core figure would make more sense as Fig. 4. We have done this change.

16. **Rev#2:** L352: The argument is reverse, the tree ring record supports the LDI data as it is a more known and used proxy.

We have changed the sentence: *"tree ring data from the Pyrenees and other Iberian areas show minor temperature variations, and even a slight temperature decrease from ~2000 to 2008 similar to the one observed in the LCD-derived temperatures from the LdRS record (Fig. 4c)."* Lines 559-562.

17. **Rev#2:** L352-354: Are the warming rate from Southern Europe/Spain also stabilizing?

We mentioned this at the end of the next section 4.3, where we deeply discussed this issue: There is a similar trend in the Pyrenees area, Iberian range and marine platforms in the western Mediterranean. Lines 672-682.

18. **Rev#2:** L433: The LDI record does not have a sufficient resolution to recognize a 1 year-long event.

Although the eruption of the mentioned volcano occurred in 1963-1964, volcanic aerosols in the atmosphere can cause decadal-timescale effects (Sigl et al., 2015). We mentioned this in the same paragraph that is referred by Rev#2. These kinds of effects could (potentially) be recorded in our record with 5-7 years of time averaging (i.e. last 180 years). However, our intention with this volcano discussion was pointing that we are not showing this direct cooling effect in our lakes (according to the correlations and figures). This is indicated in the second sentence of that paragraph, but we cannot exclude its potential influence at some specific times, such as the related with the eruption of 1963-1964.

19. **Rev#2:** L437-438: The cooling in the LDI of 1450-1500 and 1690 CE could also be attributed to solar minima rather than volcanic eruption. What about the volcanic aerosol from 1200-1300 CE that do not seem to impact the LDI in LdRS?

Actually, in these lines we mentioned that although reconstructed-LDI cold temperatures occasionally seem to occur coevally with volcanic eruptions, for example, at ~1450-1500 and 1690 CE, there is **not** a direct relationship between intensity of number of large eruptions and the reconstructed coolings in LdRS records (lines 650-652). In addition, in lines 422-423 of the former draft we mentioned that *"volcanic forcing do not show a significant correlation with LDI-derived temperatures from LdRS over the last 1500 year "* and that most of the cooling events recorded in LdRS, such as those during the LIA, were coeval with low solar activity periods (former lines 441-442): new lines 637-640. Therefore, we are discarding a strong impact of volcanic aerosols in our paleoclimate records.

**20. Rev#2:** L473: Precise the number of samples analyzed for the MCA. The MCA baseline seems to be only represented by one samples, the rest of the MCA samples are much cooler. An average temperature of all the MCA samples is a better representation of the MCA temperature and can be used as MCA baseline.

We did not intend to create a temperature baseline for the MCA, but we wanted to stablish the highest temperature recorded in a non-industrial period during the Common era. This cannot be stablished with averaged temperatures. We agree with Rev#2, and the expression MCA temperature background in figures and text is not the best choice, since it is confusing and does not reflect our discussion. We have modified it accordingly. Lines 686-687.

**21. Rev#2:** L477: Precise the number of samples analyzed for the LIA

We have also added the oldest samples of the short core (from 1820s to 1850) to this discussion; therefore, we have a total of 8 samples for the LIA. Line 698 and Fig. 7.

**22. Rev#2:** L497-498: Provide a reference for the statement: "Future scenarios are not optimistic for Sierra Nevada alpine areas either as projected temperature may rise at least _1.4 ºC by the end of the 21st century"

We have modified this paragraph (see Rev#1 comments 19 and 20), removed the potential overinterpretations, and added references.

**23. Rev#2:** L544: Is the temperature records mentioned from this study or from instrumental data, is there any precipitation reconstruction existing for this region?

Temperature data are those deduced from the LCD calibration, since long temperature records are lacking from southern Spain alpine areas. Precipitation (instrumental + reconstructed) data showed in Fig. 7b are from Rodrigo et al. (1999). We have rephrased the sentence in order to clarify so: *"Precipitation data from the southern Iberia (Rodrigo et al., 1999) along with the LCD-reconstructed temperatures in LdRS records suggest....."* Lines 744-745.

**24. Rev#2:** Fig 2b: the dashed line is almost not visible, either change colors or thickness. Add the timing of LIA and MCA to the figure. Add the temperature records from the instrumental data to help comparison.

Done. The instrumental temperature data have been added instead to current figure 5, as Rew#1 suggested, since the scale of the temperature instrumental data (100 years), would be too small for this figure (scale of 1500 years).

**25. Rev#2:** Fig S1: Correct "row" by "raw" in (a) (c) (e) (g) (i) and (k). Add unit for axis y

Done.

**26. Rev#2:** Fig S3: Please, add legend to the y axis. The authors use r and not r2 like in other figures, homogenize.

*We have removed this figure. See previous comments.*

**Cited literature**

Barea-Arco, J., Pérez-Martínez, C., Morales-Baquero, R., 2001. Evidence of a mutualistic relationship between an algal epibiont and its host, Daphnia pulicaria. Limnology and Oceanography 46, 871-881.

Garcia-Alix, A., Jimenez Espejo, F.J., Toney, J.L., Jiménez-Moreno, G., Ramos-Román, M.J., Anderson, R.S., Ruano, P., Queralt, I., Delgado Huertas, A., Kuroda, J., 2017. Alpine bogs of southern Spain show human-induced environmental change superimposed on long-term natural variations. Scientific Reports 7, 7439

García Montoro, C., Titos Martínez, M., Casado Sánchez de Castilla, M., 2016. Sierra Nevada. Una expedición al pico de Veleta desde los Baños de Lanjarón (1859). Universidad de Granada, Editorial Universidad de Granada.

Jiménez, L., Romero-Viana, L., Conde-Porcuna, J.M., Pérez-Martínez, C., 2015. Sedimentary photosynthetic pigments as indicators of climate and watershed perturbations in an alpine lake in southern Spain. Limniteca 34, 439-454.

Menéndez, R., González-Megías, A., Jay-Robert, P., Marquéz-Ferrando, R., 2014. Climate change and elevational range shifts: evidence from dung beetles in two European mountain ranges. Global Ecology and Biogeography 23, 646-657.

Rampen, S.W., Datema, M., Rodrigo-Gámiz, M., Schouten, S., Reichart, G.-J., Sinninghe Damsté, J.S., 2014a. Sources and proxy potential of long chain alkyl diols in lacustrine environments. Geochimica et Cosmochimica Acta 144, 59-71.

Rampen, S.W., Schouten, S., Koning, E., Brummer, G.-J.A., Sinninghe Damsté, J.S., 2008. A 90 kyr upwelling record from the northwestern Indian Ocean using a novel long-chain diol index. Earth and Planetary Science Letters 276, 207-213.

Rampen, S.W., Willmott, V., Kim, J.-H., Rodrigo-Gámiz, M., Uliana, E., Mollenhauer, G., Schefuß, E., Sinninghe Damsté, J.S., Schouten, S., 2014b. Evaluation of long chain 1,14-alkyl diols in marine sediments as indicators for upwelling and temperature. Organic Geochemistry 76, 39-47.

Rampen, S.W., Willmott, V., Kim, J.-H., Uliana, E., Mollenhauer, G., Schefuß, E., Sinninghe Damsté, J.S., Schouten, S., 2012. Long chain 1,13- and 1,15-diols as a potential proxy for palaeotemperature reconstruction. Geochimica et Cosmochimica Acta 84, 204-216.

Rodrigo, F.S., Esteban-Parra, M.J., Pozo-Vázquez, D., Castro-Díez, Y., 1999. A 500-year precipitation record in Southern Spain. International Journal of Climatology 19, 1233-1253.

Rodrigo-Gámiz, M., Martínez-Ruiz, F., Rampen, S.W., Schouten, S., Sinninghe Damsté, J.S., 2014. Sea surface temperature variations in the western Mediterranean Sea over the last 20 kyr: A dual-organic proxy (UK′37 and LDI) approach. Paleoceanography 29, 87-98.

Rodrigo-Gámiz, M., Rampen, S.W., de Haas, H., Baas, M., Schouten, S., Sinninghe Damsté, J.S., 2015. Constraints on the applicability of the organic temperature

proxies UK'37, TEX86 and LDI in the subpolar region around Iceland. Biogeosciences 12, 6573-6590.

Romero-Viana, L., Kienel, U., Sachse, D., 2012. Lipid biomarker signatures in a hypersaline lake on Isabel Island (Eastern Pacific) as a proxy for past rainfall anomaly (1942–2006AD). Palaeogeography, Palaeoclimatology, Palaeoecology 350-352, 49-61.

Sánchez-Castillo, P.M., 1988. Algas de las lagunas de alta montaña de Sierra Nevada (Granada, España). Acta Botánica Malacitana 13, 21 -52.

Sánchez-Castillo, P.M., Cruz-Pizarro, L., Carrillo, P., 1989. Caracterización del fitoplancton de las lagunas de alta montaña de Sierra Nevada (Granada, Spain) en relación con las características físico-químicas del medio. Limnetica 5, 37-50

Shimokawara, M., Nishimura, M., Matsuda, T., Akiyama, N., Kawai, T., 2010. Bound forms, compositional features, major sources and diagenesis of long chain, alkyl mid-chain diols in Lake Baikal sediments over the past 28,000 years. Organic Geochemistry 41, 753-766.

Sigl, M., Winstrup, M., McConnell, J.R., Welten, K.C., Plunkett, G., Ludlow, F., Buntgen, U., Caffee, M., Chellman, N., Dahl-Jensen, D., Fischer, H., Kipfstuhl, S., Kostick, C., Maselli, O.J., Mekhaldi, F., Mulvaney, R., Muscheler, R., Pasteris, D.R., Pilcher, J.R., Salzer, M., Schupbach, S., Steffensen, J.P., Vinther, B.M., Woodruff, T.E., 2015. Timing and climate forcing of volcanic eruptions for the past 2,500 years. Nature 523, 543-549.

Titos Martínez, M., 2019. Los trabajos de desagüe de las lagunas de Sierra Nevada: un largo despropósito medioambiental. Revista del Centro de Estudios Históricos de Granada y su Reino, 223-243.

Titos Martínez, M., Ramos Lafuente, A.J., 2016. El refugio más antiguo de Sierra Nevada: Construido en 1891, aún se mantiene en pie. Andalucía en la historia, 48-53.

---

## Author Response (AR2)

Granada, 20th December 2019

Dear Dr. McClymont,

Thank you very much for your positive comments. We have made all the changes you suggested in the PDF (red font in the new version of the manuscript), and we have submitted our raw data to the Pangaea website in order to store them properly. Below we answer to the most important points of your minor revisions:

1. Title: After discussing with the coauthors, we finally conclude that a good title would be: *Algal lipids reveal unprecedented warming rates in alpine areas of SW Europe during the Industrial Period.*

2. Point 1 of editor comments: Extrapolation. We totally agree with your concerns. So, we have added a short paragraph at the end of the section 3.2 (results: where we described the reconstruction of the LCD-derived temperatures): "*The application of the obtained calibration to the LDI values of LdRS (Eq. 2) produced the first temperature reconstruction for the Common Era in this alpine area. Nevertheless, a potential challenge of using this kind of down-core proxy calibrations is that the uncertainty of the reconstructed variables (temperature in this case) would increase when data fall outside the calibration data-set (e.g., during the LIA). Further studies on the local LCD production in this alpine area will contribute to extend the range of temperatures in the calibration, reducing the uncertainties of the LCD-derived temperatures.*" We think that this is the best location for this observation (result section 3.2: lines 478-486) since we explain the obtained results and their potential uncertainties.

3. Point 2 of editor comments: Fig. 5 smooth comments. We are aware of other different simulations to compare time-series such as the running/moving means. The problem we found regarding the application of running means in our dataset were: 1) the time-averaging is not always the same in our record (from 5 to 7 years in the short core); so, the best approach using running means would consider both endmembers, 5 and 7-yr running means; and 2) the graphical results of running means depends on the constrains you use, e.g., running means from the top of the record, or from the bottom (see Fig. 1). Therefore, we can have the same problems of non-alignment with the original data as the ones noted by Rev#1 in our simulation using the specific time-averaging of LdRS shc (see 7-yr running mean of TSI data: Fig. 1). We have also compared the simulations obtained using 7-yr running means in some variables with the approach performed in the previous

version of the manuscript (using the same time-averaging as LdRS shc) and the correlations were certainly similar (see some examples in Table 1). In addition, if we compare the 7-yr running mean simulation for the TSI data and the one obtained with the same time-averaging as LdRS shc, both simulations are certainly similar (r=0.98 p<0.0001) (Fig 2). So, we would prefer our approach since it includes the different changes in the time-averaging (5, 6, and 7 years), as in the studied record. It does not mean that we are putting a lot of emphasis on our age model accuracy and precision, since continuous data from 2008 to 1821 are compared in both approaches (the 7-year running means and our simulation) eventually. Anyway, if Editor prefers a 7-yr running mean simulation for this figure, we can include so, but it will not show a big difference respect to our simulation using the same time-averaging as LdRS shc.

[Figure]

**Figure 1.** Comparison between TSI data using the same time-averaging as LdRS short core and 7-year running mean simulations from the top and the bottom of the time-series. Variables: Total Solar Irradiance (TSI) and LDI record of LdRS short core. Original data in black.

| LDI vs. | 7-yr running mean | | Our model: LdRS time-averaging | |
|---|---|---|---|---|
| | r | p | r | p |
| **TSI** | 0.64 | <0.001 | 0.56 | <0.001 |
| **CH4** | 0.83 | <0.001 | 0.86 | <0.001 |
| **CPS Summer temperatures** | 0.59 | <0.001 | 0.58 | <0.001 |
| **NAO** | -0.10 | 0.1792 | -0.03 | 0.883 |
| **$U^{K'}_{37}$-SST Gol-Ho1B** | 0.70 | <0.001 | 0.76 | <0.001 |
| **AMO** | 0.57 | <0.001 | 0.61 | <0.001 |

**Table 1.** Comparison between the correlations of LDI vs. other variables using a 7-yr running mean simulation and the same time-averaging as LdRS short core.

[Figure]

**Figure 2.** TSI data comparison between a 7-year running mean simulation from the top of the time-series and another simulation using the same time-averaging as LdRS short core for the Total Solar Irradiance (TSI). Original TSI data in black dots.